# Realization of vertical metal semiconductor heterostructures via solution phase epitaxy

Xiaoshan Wang[1], Zhiwei Wang[1], Jindong Zhang[1], Xiang Wang[1], Zhipeng Zhang[1], Jialiang Wang[1], Zhaohua Zhu[1], Zhuoyao Li[1], Yao Liu[1], Xuefeng Hu[2], Junwen Qiu[2], Guohua Hu [3], Bo Chen[4], Ning Wang[1,4], Qiyuan He[4], Junze Chen[4], Jiaxu Yan[1], Wei Zhang[2], Tawfique Hasan[3], Shaozhou Li[5], Hai Li [1], Hua Zhang [4], Qiang Wang[6], Xiao Huang [1] & Wei Huang[1,5,7]

The creation of crystal phase heterostructures of transition metal chalcogenides, e.g., the 1T/2H heterostructures, has led to the formation of metal/semiconductor junctions with low potential barriers. Very differently, post-transition metal chalcogenides are semiconductors regardless of their phases. Herein, we report, based on experimental and simulation results, that alloying between $1T\text{-}SnS_2$ and $1T\text{-}WS_2$ induces a charge redistribution in Sn and W to realize metallic $Sn_{0.5}W_{0.5}S_2$ nanosheets. These nanosheets are epitaxially deposited on surfaces of semiconducting $SnS_2$ nanoplates to form vertical heterostructures. The ohmic-like contact formed at the $Sn_{0.5}W_{0.5}S_2/SnS_2$ heterointerface affords rapid transport of charge carriers, and allows for the fabrication of fast photodetectors. Such facile charge transfer, combined with a high surface affinity for acetone molecules, further enables their use as highly selective 100 ppb level acetone sensors. Our work suggests that combining compositional and structural control in solution-phase epitaxy holds promises for solution-processible thin-film optoelectronics and sensors.

[1] Institute of Advanced Materials (IAM), Nanjing Tech University (NanjingTech), 30 South Puzhu Road, Nanjing 211816, China. [2] State Key Laboratory of Materials-Oriented Chemical Engineering, College of Chemical Engineering, Nanjing Tech University (NanjingTech), 30 South Puzhu Road, Nanjing 211816, China. [3] Cambridge Graphene Centre, University of Cambridge, Cambridge CB3 0FA, UK. [4] Center for Programmable Materials, School of Materials Science and Engineering, Nanyang Technological University, 50 Nanyang Avenue, Singapore 639798, Singapore. [5] Key Laboratory for Organic Electronics and Information Displays & Institute of Advanced Materials, Jiangsu National Synergistic Innovation Center for Advanced Materials (SICAM), Nanjing University of Posts & Telecommunications, 9 Wenyuan Road, Nanjing 210023, China. [6] School of Chemistry and Molecular Engineering, Nanjing Tech University (NanjingTech), 30 South Puzhu Road, Nanjing 211816, China. [7] Shaanxi Institute of Flexible Electronics (SIFE), Northwestern Polytechnical University (NPU), 127 West Youyi Road, Xi'an 710072, China. These authors contributed equally: Xiaoshan Wang, Zhiwei Wang. Correspondence and requests for materials should be addressed to Q.W. (email: wangqiang@njtech.edu.cn) or to X.H. (email: iamxhuang@njtech.edu.cn) or to W.H. (email: iamwhuang@nwpu.edu.cn)

Heterostructures constructed from layered materials such as graphene, metal chalcogenides and black phosphorus (BP) have aroused particular interest due to their combined advantageous features and the emergence of unusual properties/functions[1–6]. As a group of the mostly studied layered materials, transition metal chalcogenides (TMCs) like $MoS_2$ and $WS_2$ exist in different crystal phases such as the 2H and 1T polytypes with distinct electronic properties. Their semiconductor-to-metal transition (i.e., 2H-to-1T) can be realized via Li-intercalation,[7] strain engineering,[8] e-beam/laser irradiation[2] or doping/alloying[2,7–9]. This attractive feature allows for the formation of 2H/1T (semiconductor/metal) phase junctions that exhibit much reduced contact resistance compared to that of using noble metal contacts (e.g., Au) which normally interface with semiconducting TMCs with large Fermi level misalignment[2,10]. Besides TMCs, post-TMCs such as $SnS_2$ and InSe are another important group of layered materials, exhibiting attractive electronic and optoelectronic properties for a wide range of applications, including transistors, photodetectors, and sensors[11,12]. Similar to many other semiconducting materials, the type of metal contact with post-TMCs plays a critical role in tuning their functional performance[13]. However, due to the less metallic nature of post-transition metals, post-TMCs are generally semiconductors regardless of their crystal phases, such as the 2H, 4H, and 1T polytypes[14]. Consequently, contacting post-TMCs with metallic layered materials to achieve low interfacial resistance remains a big challenge.

To date, much effort has been devoted to the preparation of various heterostructures based on layered materials, which display different geometric arrangements such as the lateral and vertical heterostructures[1,3,15]. In particular, to prepare vertical heterostructures in which dissimilar layered crystals are stacked one above the other in a pre-designed sequence, solid-state procedures, including dry transfer, chemical vapor deposition (CVD) and chemical vapor transport (CVT) method have mostly been applied[3,16,17]. This is because these methods allow good control over the spatial arrangement of the layers as well as the deposition sequence. Compared to these solid-state methods, solution-phased approaches are advantageous in terms of relatively easier procedures, low-cost setups, and most importantly, scalability[18,19]. However, direct wet-chemical growth of vertical heterostructures of layered metal chalcogenides has thus far been challenging.

In view of their potential applications, heterostructures/heterojunctions such as InSe/graphene, $MoTe_2$/$MoS_2$, $MoS_2$/perovskite, and graphene/$MoS_2$/graphene have recently shown promising performance in photodetectors, due to the improved charge separation/transport and enhanced light adsorption[20–23]. Meanwhile, development of gas sensors for detection of volatile organic compounds are important in applications such as environmental monitoring and non-invasive diagnosis of diseases based on breath analysis[24,25]. Chemiresistive sensors based on metal oxides/sulfides have been used for detection of volatile organic compounds, however, high operating temperatures (typically ≥150 °C) are usually required to achieve good sensitivity and selectivity[24,26]. Very recently, layered materials such as $SnS_2$, $WS_2$ and $Ti_3C_2T_x$ have demonstrated great potential for room-temperature gas detection[27–29]. It is expected that creation of heterostructures may realize further improved sensing performance[30–32].

In this contribution, nanoplates of $SnS_2$, a typical n-type semiconductor, are used as synthesis templates for the surface deposition of layered $Sn_{0.5}W_{0.5}S_2$ nanosheets, which show 83% metallic phase, leading to the formation of metal/semiconductor vertical heterostructures. Kelvin probe force microscope (KPFM) and tunneling atomic force microscopy (TUNA) analyses suggest the formation of ohmic-like contact at the $Sn_{0.5}W_{0.5}S_2$/$SnS_2$ interface. The resultant heterostructures are fabricated into chemiresistive sensors to detect acetone at room temperature and exhibit a good selectivity and a minimum detectable concentration down to 100 ppb. The good sensing performance could be attributed to the low charge transfer resistance at the $Sn_{0.5}W_{0.5}S_2$/$SnS_2$ interface that enables a much increased signal-to-noise ratio, and the alloying induced enhancement in surface gas adsorption.

## Results

**Synthesis and characterizations of $Sn_{0.5}W_{0.5}S_2$/$SnS_2$.** As a representative post-TMC, $SnS_2$ has been widely studied and applied in phototransistors and sensors for its favorable band structure and relatively high surface electronegativity[12,33]. Typically, $SnS_2$ nanoplates were synthesized via a hydrothermal reaction with thiourea ($CS(NH_2)_2$) and tin tetrachloride hydrate ($SnCl_4{\cdot}5H_2O$) as the precursors for S and Sn, respectively (see the Methods section for the detailed procedure)[34]. As shown in the scanning electron microscope (SEM), transmission electron microscope (TEM) and atomic force microscopy (AFM) images in Supplementary Figs. 1 and 2, hexagonal nanoplates with edge lengths ranging from 200 to 700 nm and an average thickness of 43 nm were obtained. As confirmed by the selected area electron diffraction (SAED) and X-ray diffraction (XRD) analyses, the nanoplates are α-$SnS_2$ with a 1T structure (space group $P\bar{3}m1$), where $a = 3.65$ and $c = 5.90$ (ICSD no. 42566)[12] (Supplementary Figs. 1c and d). By adding $(NH_4)_{10}H_2(W_2O_7)_6$ to the precursors of the aforementioned synthesis solution, alloyed $Sn_{1-x}W_xS_2$ nanosheets were in-situ synthesized and hybridized with $SnS_2$ nanoplates as shown in Fig. 1a, b. These heterostructures show an average lateral size of 750 nm (Fig. 1a) and an average thickness of 60 nm (Fig. 1c and Supplementary Fig. 3). The side-view TEM image in Fig. 1d clearly shows a $SnS_2$ nanoplate covered by $Sn_{1-x}W_xS_2$ nanosheets on both its basal faces forming a vertical heterostructure. The deposited $Sn_{1-x}W_xS_2$ nanosheets are typically 6–9 nm in thickness (Supplementary Fig. 4). Energy dispersive X-ray spectroscopy (EDX) mapping of a typical heterostructure reveals the distribution of Sn, W, and S elements (Fig. 1e), in which the center of the heterostructure shows a higher concentration of Sn compared to the edge. EDX line analysis of the cross-section of a typical heterostructure, which was prepared by cutting with focused ion beam (FIB), further indicates that the $SnS_2$ nanoplate was covered by $Sn_{1-x}W_xS_2$ nanosheets (Fig. 1f). EDX spot analyses on edges of $Sn_{1-x}W_xS_2$ nanosheets suggest $x \approx 0.5$, confirming $Sn_{0.5}W_{0.5}S_2$ nanosheets were obtained (Supplementary Fig. 5).

The structural properties of the heterostructures were investigated with XRD and high resolution TEM (HRTEM) analysis. In the XRD pattern in Fig. 2a, besides the characteristic peaks for 1T-$SnS_2$, two relatively broader peaks observed at 8.9° and 17.8° could be attributed, respectively, to the (001) and (002) planes of $Sn_{0.5}W_{0.5}S_2$ nanosheets with an enlarged interlayer spacing of about 1.0 nm. The enlarged spacing, which was also observed in side-view HRTEM images (Supplementary Fig. 6), may result from the use of $CS(NH_2)_2$ and $(NH_4)_{10}H_2(W_2O_7)_6$ in our synthesis, leading to the intercalation of $NH_4^+$ ions in between the adjacent layers[35]. This was also confirmed by X-ray photoelectron spectroscopy (XPS) analysis (Supplementary Fig. 7). The crystal structure of a typical heterostructure was further investigated by taking its SAED pattern. As shown in Fig. 2b, two sets of patterns both along the [001] zone axis show the epitaxial registry. The hexagonal pattern with six inner and six outer spots can be assigned to 1T-$SnS_2$, and the measured (100) lattice spacing is 3.16 Å, in good agreement with the XRD result (Fig. 2a). The other set of pattern for $Sn_{0.5}W_{0.5}S_2$ shows elongated spots, forming discontinued ring segments. This may

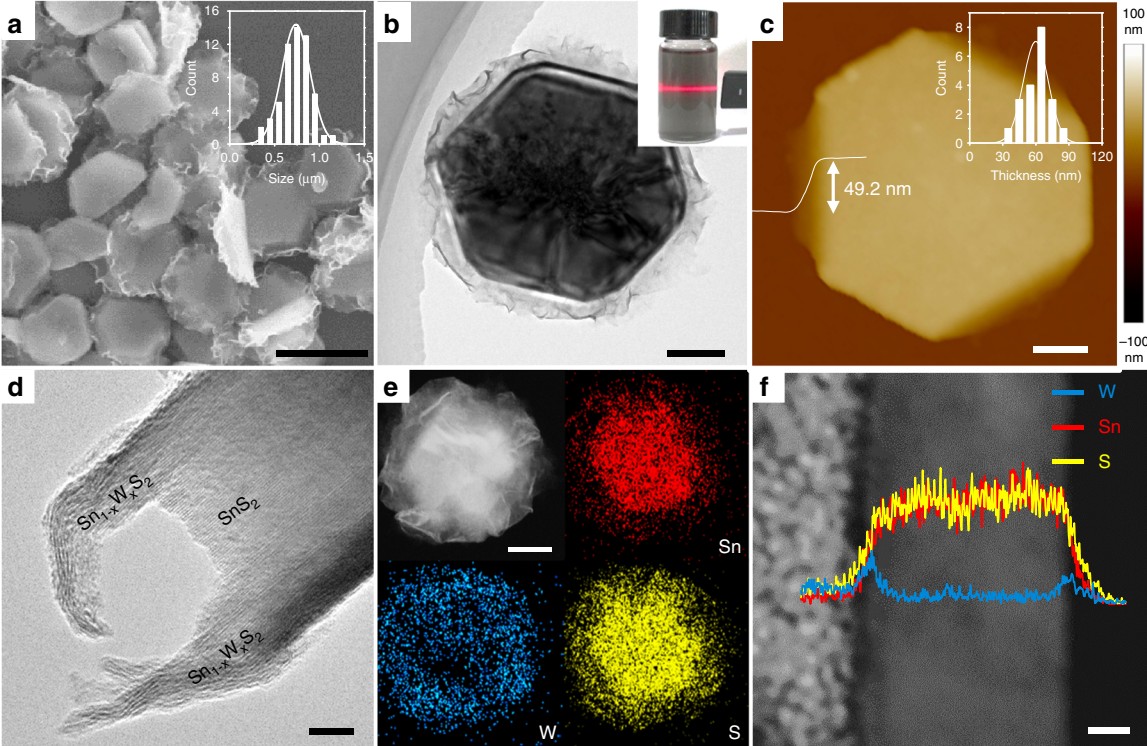

**Fig. 1** Morphology and composition analyses of $Sn_{0.5}W_{0.5}S_2/SnS_2$ heterostructures. **a** SEM image of as-prepared $Sn_{1-x}W_xS_2/SnS_2$ heterostructures (scale bar, 1 μm). Inset: size distribution of $Sn_{1-x}W_xS_2/SnS_2$ heterstructures. **b** Top-view TEM image of typical $Sn_{1-x}W_xS_2/SnS_2$ heterostructures (scale bar, 200 nm). Inset: photograph of a solution of $Sn_{1-x}W_xS_2/SnS_2$ heterostructures showing the Tyndall effect. **c** AFM image and height analysis of a $Sn_{1-x}W_xS_2/SnS_2$ heterostructure (scale bar, 100 nm). Inset: thickness distribution of $Sn_{1-x}W_xS_2/SnS_2$ heterstructures, showing a mean value of 60 nm. **d** Side-view TEM image of a typical $Sn_{1-x}W_xS_2/SnS_2$ heterostructure, revealing $Sn_{1-x}W_xS_2$ nanosheets grown on both the top and bottom basal faces of a SnS₂ nanoplate (scale bar, 10 nm). **e** STEM image and EDX mapping on a typical $Sn_{1-x}W_xS_2/SnS_2$ heterostructure (scale bar, 100 nm). **f** Cross-sectional STEM image and EDX line analysis on a typical $Sn_{1-x}W_xS_2/SnS_2$ heterostructure (scale bar, 10 nm)

result from the curled structure of the $Sn_{0.5}W_{0.5}S_2$ nanosheets as well as possible misorientation. The measured lattice spacing for $Sn_{0.5}W_{0.5}S_2$ (100) planes was 3.0 Å, corresponding to a lattice parameter of $a = 3.46$ Å. This reasonably falls in between that of WS₂ ($a = 3.16$ Å) and SnS₂ ($a = 3.65$ Å). Note that the mismatch between the (100) planes of SnS₂ and $Sn_{0.5}W_{0.5}S_2$ is 5%, which can be tolerated in van der Waals epitaxial growth of layered materials[36]. The top-view HRTEM image in Fig. 2c distinctly shows the relatively darker region for the SnS₂ nanoplate covered with $Sn_{0.5}W_{0.5}S_2$, and the brighter region for the periphery $Sn_{0.5}W_{0.5}S_2$ nanosheet. The lattice fringes extend continuously from the $SnS_2/Sn_{0.5}W_{0.5}S_2$ center to the $Sn_{0.5}W_{0.5}S_2$ edge, further confirming the epitaxial growth mode. Moiré patterns could be observed in some areas due to the overlap of $Sn_{0.5}W_{0.5}S_2$ and SnS₂ at small misorientation angles. For example, as shown in Fig. 2d, a Moiré pattern with a periodicity of 4.0 nm was observed when the $Sn_{0.5}W_{0.5}S_2$ overlayer made a misorientation angle of 3° with SnS₂ (see the detailed analysis in Supplementary Fig. 8). Note that the overlapping of two hexagonal lattice patterns normally produces a hexagonal Moiré pattern[37], which was not observed in the present work. This may be due to the fact that the $Sn_{0.5}W_{0.5}S_2$ nanosheets showed lattice distortion with varied interlayer spacings (0.6–1.0 nm, Supplementary Fig. 6), and thus were deviated from being perfectly flat on the SnS₂ nanoplate. As a result, only short-range line-like Moiré patterns were observed[38]. High resolution scanning TEM (STEM) images of the edge area of a heterostructure show the 1T-phase-like atomic arrangement of $Sn_{0.5}W_{0.5}S_2$ (Supplementary Fig. 9)[39]. It is interesting that zigzag lattice patterns for 1T' or $T_d$

phases which have been previously observed in TMCs such as WS₂ and WTe₂[40,41] were not observed in $Sn_{0.5}W_{0.5}S_2$. Based on our density functional theory (DFT) calculation results shown in Supplementary Table 1 and Supplementary Fig. 10, the W–S bonds tend to be shorter than the Sn–S bonds in the alloyed system, resulting in a distortion from the perfect in-plane symmetric 1T lattice. The distorted 1T structure was also reflected in the Raman spectrum in Supplementary Fig. 11, where in addition to the peaks at 310, 351, and 414 cm⁻¹ that correspond to the SnS₂-like $A_{1g}$, WS₂-like $E_{2g}$ and WS₂-like $A_{1g}$ modes, respectively, the active modes observed at 171 cm⁻¹ and 224 cm⁻¹ in the lower frequency region could be attributed to distorted 1T-WS₂[12,35]. The dominant distorted 1T-phased W–S coordination was further confirmed by XPS analysis (Fig. 2e). It has been reported that the XPS band positions of a metal element are sensitive to their oxidation states, coordination geometries, and Fermi levels[42,43]. Normally in the W$f$ spectrum, the doublet peaks (32.1 and 34.2 eV) associated with W in the 1T-phased structure are downshifted by 0.6 eV relative to those associated with W in the 2H structure (32.7 and 34.8 eV). The deconvolution of the W 4$f$ bands thus could enable the quantitative estimation of the 1T and 2H phase concentrations[7]. In our case, from the deconvoluted peak areas (Fig. 2e), the concentrations of the distorted 1T and 2H phases were calculated to be 83% and 17%, respectively. For the Sn 3$d$ spectrum, the peak positions match well with the 1T-SnS₂ structure (Fig. 2f)[44].

The formation of 1T or distorted 1T structures have been observed previously in TMCs when they were intercalated with alkali metal ions (e.g., Li⁺, K⁺ etc.)[39,45] or synthesized in the

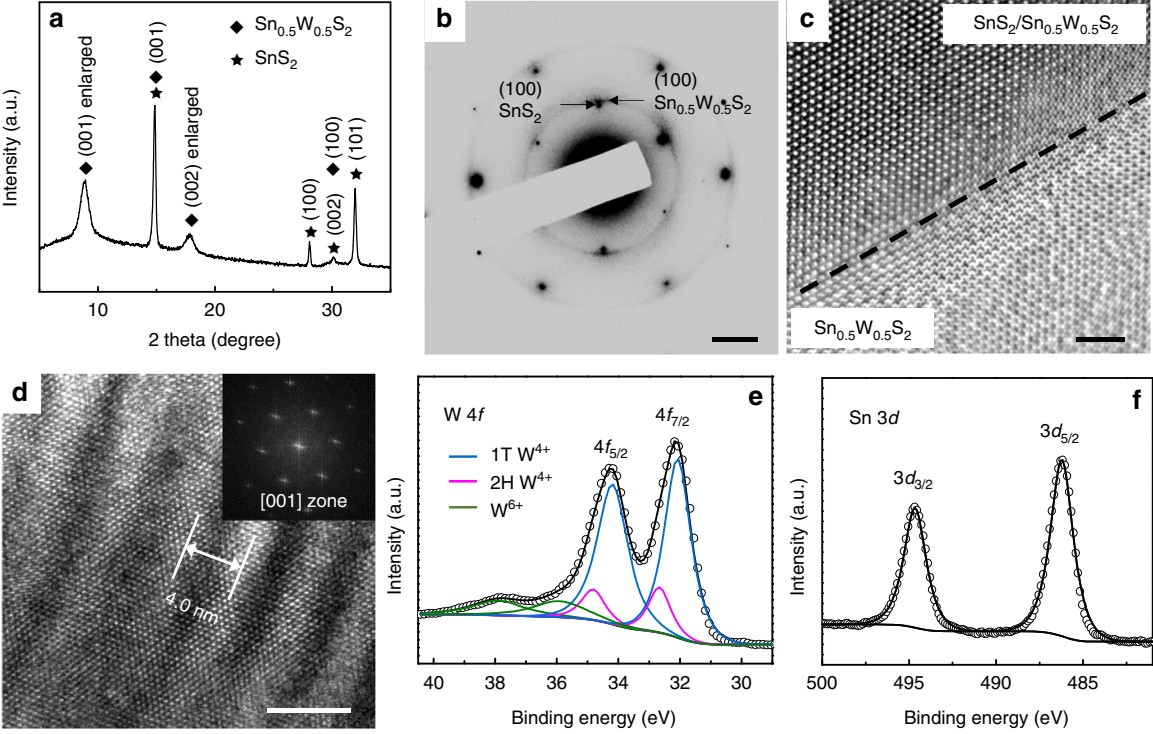

**Fig. 2** Structural properties of $Sn_{0.5}W_{0.5}S_2/SnS_2$ heterostructures. **a** XRD pattern of $Sn_{0.5}W_{0.5}S_2/SnS_2$ heterostructures deposited on a glass slide. **b** SAED patterns of a $Sn_{0.5}W_{0.5}S_2/SnS_2$ heterostructure along the [001] zone axis (scale bar, 2 nm$^{-1}$). **c** HRTEM image of a typical $Sn_{0.5}W_{0.5}S_2/SnS_2$ heterostructure lying flatly on a copper grid (scale bar, 2 nm). **d** A Moiré pattern with a periodicity of 4.0 nm was observed (scale bar, 5 nm), whose fast Fourier transform (FFT) diffraction pattern is shown as the inset. XPS (**e**) W 4$f$ and **f** Sn 3$d$ spectra of as-prepared $Sn_{0.5}W_{0.5}S_2/SnS_2$ heterostructures

presence of ammonium containing precursors[35] and hydrazine hydrate[46]. According to previous theoretical calculations, the presence of the positive counterions could cause an increase of the electron density of the $d$-orbital of the transition metals, leading to the stabilization of the 1T or distorted 1T phase[47]. Therefore, the realization of the distorted 1T-$Sn_{0.5}W_{0.5}S_2$ in our present work might also be a result of the intercalated $NH_4^+$ cations from $CS(NH_2)_2$ and $(NH_4)_{10}H_2(W_2O_7)_6$ used in the synthesis solution.

**Formation process of $Sn_{0.5}W_{0.5}S_2/SnS_2$.** To investigate the formation process of the $Sn_{0.5}W_{0.5}S_2/SnS_2$ vertical heterostructures, intermediate products were collected at different reaction intervals for characterizations. At the beginning of the synthesis process, the precursors, i.e., $(NH_4)_{10}H_2(W_2O_7)_6$ and $SnCl_4 \cdot 5H_2O$, reacted to produce $Sn(HWO_4)_2 \cdot nH_2O$ amorphous particles upon mixing at 80 °C (Supplementary Fig. 12)[48], which were subsequently heated up to 220 °C under hydrothermal conditions. After the reaction had proceeded for 12 h at 220 °C, nanorods with lengths of 10–100 nm were observed together with $SnS_2$ nanoplates in the solution (Fig. 3a, b). These nanorods are alloyed oxide of Sn and W with a formula of $Sn_{0.17}WO_3$ based on XRD pattern (ICSD No. 38043 [https://icsd.fiz-karlsruhe.de], Fig. 3c)[49], EDX analysis (Supplementary Fig. 13a, b) and HRTEM imaging (Supplementary Fig. 13c). The reason why $WS_2$ was not produced at this stage is that the bond energy of Sn–S was likely to be lower than that of W–S due to the larger ionic radius of $Sn^{4+}$ compared to that of $W^{4+}$[50,51]. As the reaction proceeded further, the amount of the $Sn_{0.17}WO_3$ nanorods decreased, and nanosheets started to form on the surfaces of the $SnS_2$ nanoplates (Supplementary Fig. 14). Evidently, the evolution of the XRD patterns of the intermediate products indicates a decrease of the $Sn_{0.17}WO_3$ amount over time (Fig. 3c), which is accompanied with an

increase of the Sn and W ion concentration in the solution based on the inductively coupled plasma mass spectrometry (ICP-MS) measurements (Supplementary Table 2). Based on our control experiments, $Sn_{0.17}WO_3$ nanorods could form at 180 °C and decompose at temperatures above 200 °C (Supplementary Fig. 15). This suggests that the $Sn_{0.17}WO_3$ nanorods forming at the beginning of the hydrothermal reaction gradually decomposed at 220 °C (step 1 in Fig. 3d), providing additional Sn and W ions with a high W/Sn ratio (>30) (Supplementary Table 2). Such a high W/Sn ratio could drive the growth of alloyed $Sn_{0.5}W_{0.5}S_2$ nanosheets on the surface of $SnS_2$ even though the formation of W–S bond is less favored compared to that of the Sn–S bond (step 2 in Fig. 3d). In addition, we also tried to extend the Sn–W binary system to Sn–Mo system, but found that, under certain conditions, $Sn_{1-x}Mo_xS_2$ nanosheets grew epitaxially on $SnS_2$ mainly via the edge growth. This phenomenon might be due to the different synthesis energies required for basal growth or edge growth[3], which requires our further investigation.

**Electronic properties of $Sn_{0.5}W_{0.5}S_2/SnS_2$.** Alloying has been a powerful approach to tune the bandgaps of TMCs or to achieve their semiconductor-to-metal transitions[9,52]. However, alloying between layered TMC and post-TMC has not been reported so far, and thus the electronic properties of such alloys could be of great interest for potential applications. DFT calculations were performed to understand the electronic properties of the distorted 1T-$Sn_{0.5}W_{0.5}S_2$ deposited on 1T-$SnS_2$ with and without $NH_4^+$ intercalation. The detailed results of the optimized crystal structures, calculated Bader charges, band structures and density of states (DOS) are shown in Fig. 4, Supplementary Figs 16 and 17 and Supplementary Table 1. In contrast to a five-layer semiconducting 1T-$SnS_2$ which shows no DOS at the Fermi level (Fig. 4b), a four-layer distorted 1T-$Sn_{0.5}W_{0.5}S_2$ on a monolayer

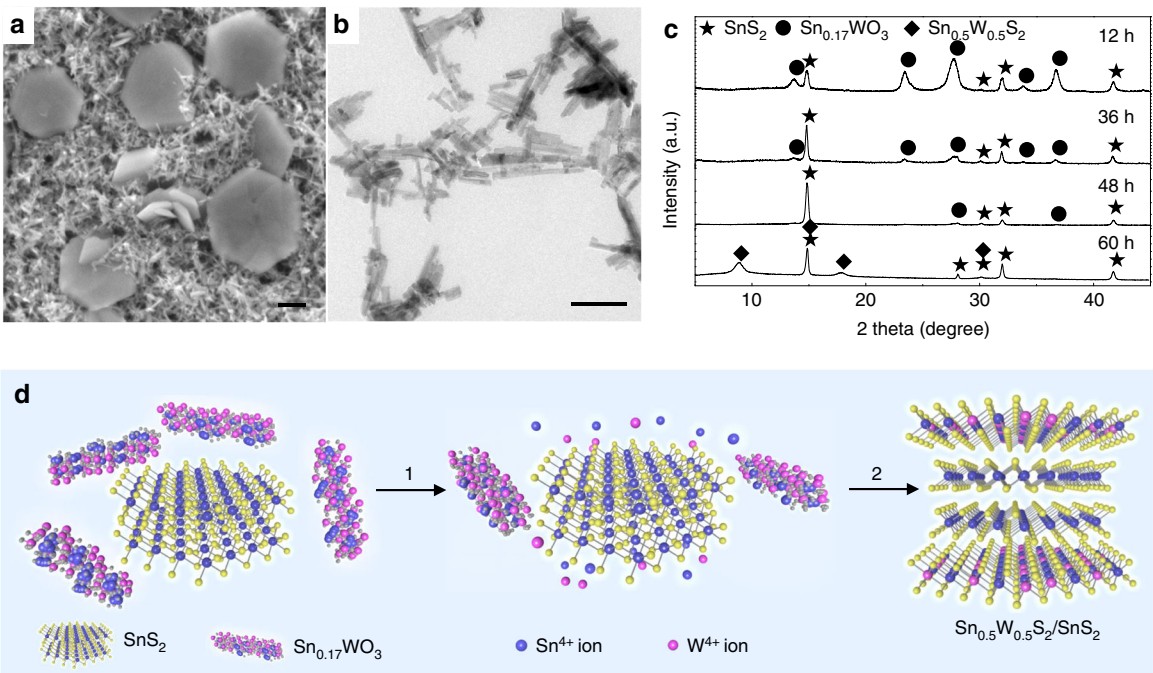

**Fig. 3** Formation process of $Sn_{0.5}W_{0.5}S_2/SnS_2$ heterostructures. **a** SEM image of the intermediate product obtained after the reaction proceeded for 12 h (scale bar, 200 nm). **b** TEM image of typical $Sn_{0.17}WO_3$ nanorods (scale bar, 100 nm). **c** XRD patterns of the intermediate products obtained at different reaction intervals. **d** Schematic illustration of the formation process of $Sn_{0.5}W_{0.5}S_2/SnS_2$ heterostructures. $SnS_2$ nanoplates and $Sn_{0.17}WO_3$ nanorods formed together during the initial 12 h. After that the $Sn_{0.17}WO_3$ nanorods began to decompose (step 1), providing additional W and Sn ions for $Sn_{0.5}W_{0.5}S_2$ nanosheets to grow on the surfaces of the $SnS_2$ nanoplates (step 2)

1T-$SnS_2$ exhibits intrinsic metallic behavior with observable DOS at the Fermi level, dominantly contributed from the W and S atoms and slightly from the Sn atoms (Fig. 4d). The calculated Bader charge of W atoms in distorted 1T-$Sn_{0.5}W_{0.5}S_2$ is 0.20 e higher compared to that in 1T-$WS_2$, whereas the Bader charge of Sn atoms in distorted 1T-$Sn_{0.5}W_{0.5}S_2$ is 0.10 e lower compared to that in 1T-$SnS_2$ (Fig. 4a, c and Supplementary Fig. 16). This suggests a charge redistribution in the $Sn_{0.5}W_{0.5}S_2$ alloy by charge transfer from W to Sn atoms[53]. Similarly, the distorted 1T-$Sn_{0.5}W_{0.5}S_2$ with intercalated $NH_4^+$ ions (20 mol%) also shows the metallic behavior (Supplementary Fig. 17). To experimentally verify the calculated results, the electronic properties of the $SnS_2$ nanoplates and $Sn_{0.5}W_{0.5}S_2/SnS_2$ heterostructures were measured by fabrication of back-gated thin film field effect transistors (Supplementary Fig. 18). The drain current ($I_d$) vs. drain-source voltage ($V_{ds}$) curves at varied gate voltages reveal that the $SnS_2$ nanoplates are typical n-type semiconductors (Supplementary Fig. 18a, b)[12]. In contrast, the $I–V$ curves of films prepared from $Sn_{0.5}W_{0.5}S_2/SnS_2$ heterostructures are almost insensitive to gate voltages (Supplementary Fig. 18c, d), suggesting the metallic charge transport through the $Sn_{0.5}W_{0.5}S_2$ nanosheets, consistent with the theoretical predictions.

To exmamine the interface property between a $SnS_2$ nanoplate and the surface deposited $Sn_{0.5}W_{0.5}S_2$, its surface potential, which corelates to its work function, was analyzed by KPFM in air[21]. A PtIr tip was used as the probe, and a $SiO_2/Si$ substrate sputtered with Au/Cr with a theoretical work function ($\phi_{Au}$) of 5.100 eV was used as the potential reference. The 2D potential image of a typical $Sn_{0.5}W_{0.5}S_2/SnS_2$ heterostructure is shown in Fig. 5a, where the color variation reflects the local surface potential difference ($\Delta V = \phi_{Au} - \phi_{sample}$) (details on the calculation of the surface potentials are given in the Methods section). It can be seen that the surface potential of the $Sn_{0.5}W_{0.5}S_2/SnS_2$ heterostructure is 0.029 V lower than that of Au to give an estimated work function of 5.071 eV (Fig. 5a). On the other hand, an $SnS_2$ nanoplate exhibits a work function of 5.110 eV (Fig. 5b). The decrease in surface potential after deposition of $Sn_{0.5}W_{0.5}S_2$ on $SnS_2$ suggests that the metallic $Sn_{0.5}W_{0.5}S_2$ possesses a lower work function than that of the n-type $SnS_2$. Although based on DFT calcuations, the work function of distorted 1T-$Sn_{0.5}W_{0.5}S_2$ is 5.62 eV which is much higher than that of $SnS_2$, it can be substantially lowered to 2.87 eV by introducing an $NH_4^+$ intercalation with a molar concentration of about 20% (Supplementary Table 1). Such work function modulation induced by doping or chemical absorbates has also been reported previously[54]. This also explains why the experimentally measured work function of $Sn_{0.5}W_{0.5}S_2$ nanosheets with partial $NH_4^+$ intercalation was lower than that of $SnS_2$. Therefore, an ohmic contact should form at the $Sn_{0.5}W_{0.5}S_2/SnS_2$ heterointerface (Fig. 5c), affording a low charge transfer resistance. This was further confirmed by measuring $I–V$ curves on individual $SnS_2$ nanoplates or $Sn_{0.5}W_{0.5}S_2/SnS_2$ heterostructures deposited on highly oriented pyrolytic graphite (HOPG) with TUNA (Fig. 5d, e). The $I–V$ curve for a $SnS_2$ nanoplate is highly asymmetric and the onset of the current rectification is at 3.50 V (Fig. 5d), indicating the presence of Schottky barrier at the PtIr tip/n-type $SnS_2$ interface, provided that the work function for PtIr, HOPG, and $SnS_2$ is 5.50, 4.60, and 5.11 eV, respectively[55,56]. In sharp contrast, the $I–V$ curve for $Sn_{0.5}W_{0.5}S_2/SnS_2$ is almost linear and symmetric with respect to 0 V, suggesting that the contact at $SnS_2/Sn_{0.5}W_{0.5}S_2$ interface is ohmic-like (Fig. 5e)[57].

**$Sn_{0.5}W_{0.5}S_2/SnS_2$ for photodetectors**. The advantage of the facile charge transport across the ohmic-like heterointerfaces was demonstrated in fabrication of thin film photodetectors based on the $Sn_{0.5}W_{0.5}S_2/SnS_2$ heterostructures. Figure 6a shows the $I–V$ curves of the device under 405 nm laser illumination with power

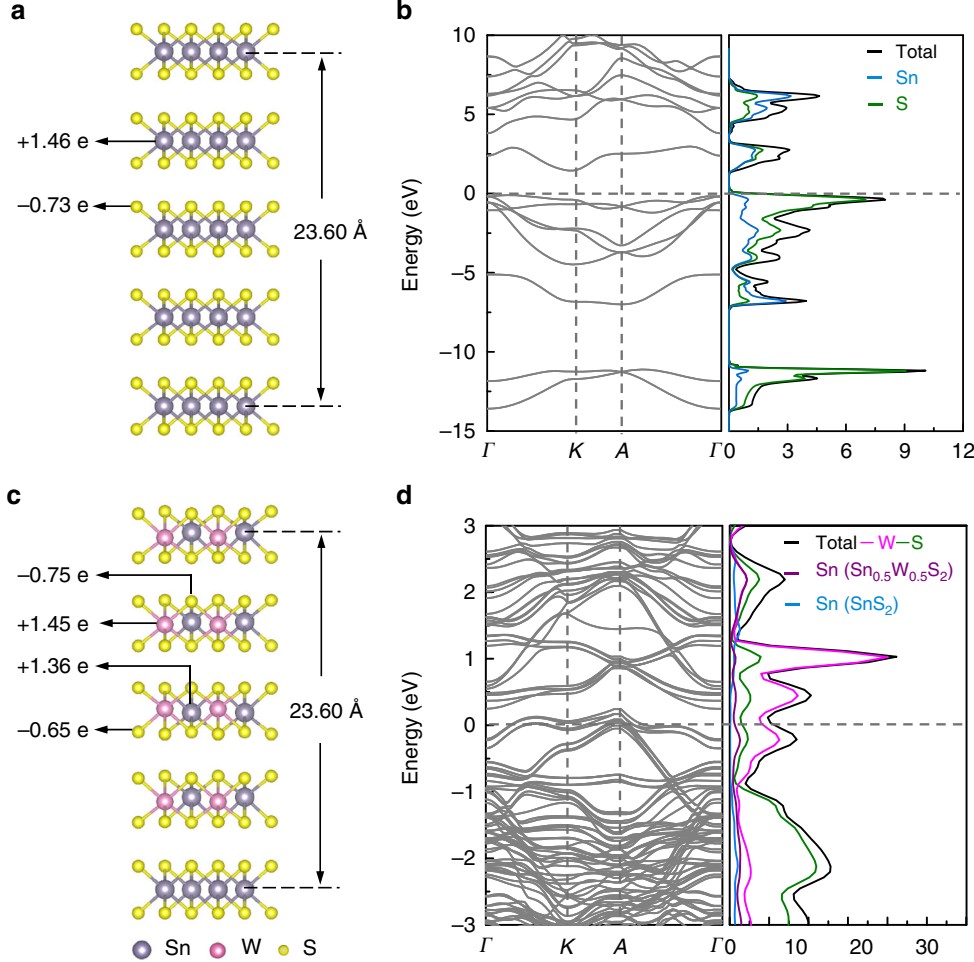

**Fig. 4** Calculated electronic structure of $SnS_2$ and $Sn_{0.5}W_{0.5}S_2/SnS_2$ heterostructures. **a** Optimized crystal structure with calculated Bader charges. **b** Band structure and DOS of a 5-layer $SnS_2$, showing an indirect band gap near the Fermi level. **c** Optimized crystal structure with calculated Bader charges. **d** Band structure and DOS of a four-layer $1T$-$Sn_{0.5}W_{0.5}S_2$ on a monolayer $1T$-$SnS_2$, showing the intrinsic metallic characteristic. The Fermi level is assigned at 0 eV

intensity varied from 0.45 to 1.05 mW. A clear rise of the photocurrent with increasing illumination intensity was observed, indicating the effective conversion of photon flux to photogenerated carriers. In addition, the $Sn_{0.5}W_{0.5}S_2/SnS_2$ photodetector showed symmetric and linear $I$–$V$ plots, which is in sharp contrast to the non-linear $I$–$V$ curves observed for $SnS_2$-based device (Supplementary Fig. 19a). This further indicates the low resistance contact formed between the semiconducting and metallic components in $Sn_{0.5}W_{0.5}S_2/SnS_2$. The temporal photoresponse of the photodetectors was measured as well as shown in Fig. 6b, c and Supplementary Figs. 19b, c. The $Sn_{0.5}W_{0.5}S_2/SnS_2$ photodetector showed an abrupt rise of photocurrent with a fast response time of 42.1 ms (defined as the time required to increase 90% from the minimum to maximum current density), which is comparable and outperforms some previously reported TMC based photodetectors[58,59]. This value is also about 50 times shorter than that of the $SnS_2$-based photodetector (2.10 s) (Supplementary Fig. 19c). Such markedly shortened response time suggests the rapid transport of charge carriers across the $Sn_{0.5}W_{0.5}S_2/SnS_2$ heterointerfaces[60,61]. Note that a relatively large dark current and thus a much reduced on/off ratio were observed for the $Sn_{0.5}W_{0.5}S_2/SnS_2$-based device as compared with the $SnS_2$ device. This was due to the metallic nature of the $Sn_{0.5}W_{0.5}S_2$ nanosheets. The similar phenomenon was reported previously in photodetectors based on graphene composites[62,63].

**$Sn_{0.5}W_{0.5}S_2/SnS_2$ for gas sensing**. The presence of a metallic component in a thin film channel may pose some limitation to its optoelectronic performance, such as the relatively large dark current observed in the aforementioned $Sn_{0.5}W_{0.5}S_2/SnS_2$ photodetector. However, the metallic structure might become an advantage for applications like sensors. As another proof of concept demonstration of the advantage of solution-processible functional materials, the $Sn_{0.5}W_{0.5}S_2/SnS_2$ heterostructures were deposited on Au interdigitated electrodes to fabricate chemiresistive gas sensors for detection of volatile organic compounds such as acetone, which is a potential biomarker for diabetes and lung cancer[64]. For comparison, $SnS_2$-based gas sensors were also fabricated. The response–recovery curves of the gas sensors were measured under gas flows with increasing concentrations (typically 0.1–50 ppm) at room temperature (Fig. 7a, b and Supplementary Figs. 20 and 21). Taking sensing of acetone for example, the resistance of the $Sn_{0.5}W_{0.5}S_2/SnS_2$ sensor decreased upon exposure to acetone and the decrease in resistance ($\Delta R = R_a - R_0$) with increasing acetone concentration (Fig. 7a, b and Supplementary Fig. 20). A minimum detectable concentration of 0.1 ppm was achieved. This is 20 times lower compared to that of the sensor based on $SnS_2$ which only afforded a minimum detectable concentration of 2 ppm (Supplementary Fig. 21), and to the best of our knowledge, outperforms other reported metal sulfide/oxide chemiresistive acetone sensors. More importantly, our sensor

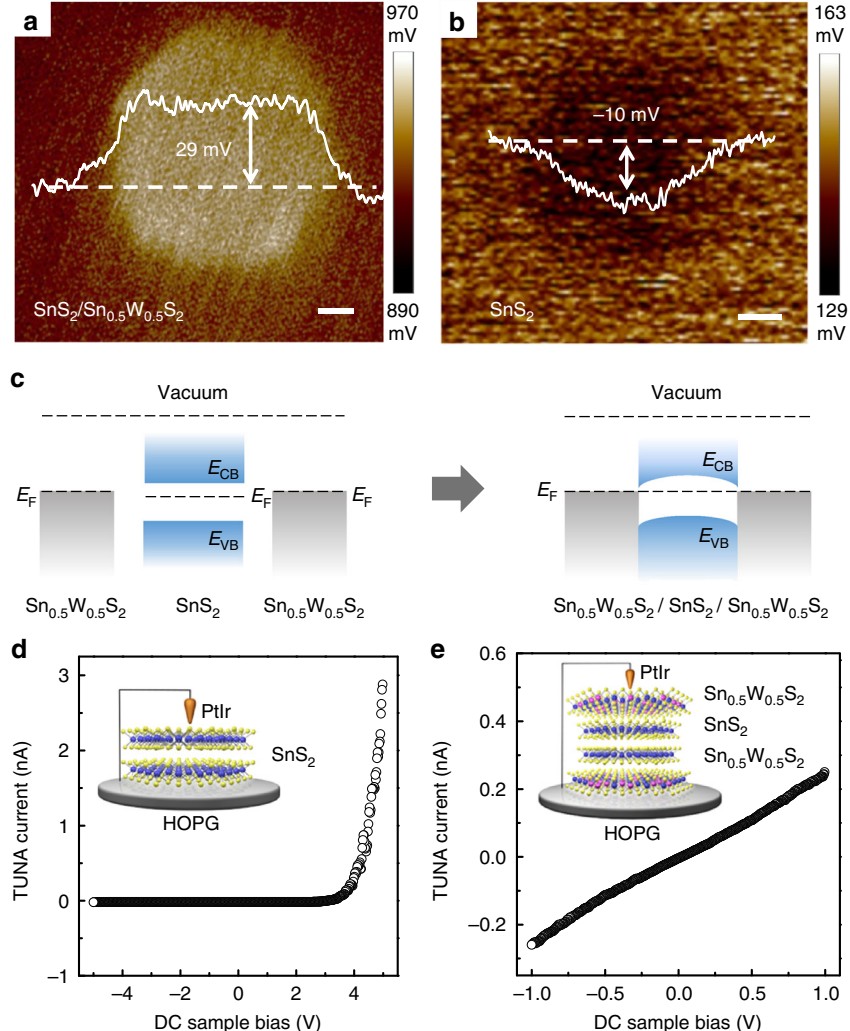

**Fig. 5** KPFM and TUNA analyses of $SnS_2$ and $Sn_{0.5}W_{0.5}S_2/SnS_2$ heterostructures. 2D potential images of **a** a typical $Sn_{0.5}W_{0.5}S_2/SnS_2$ heterostructure (scale bar, 200 nm) and **b** a typical $SnS_2$ nanoplate deposited on a $SiO_2/Si$ substrate coated with a thin film of Au/Cr (i.e., $Au/Cr/SiO_2/Si$) (scale bar, 100 nm). **c** Schematic band alignment diagram for $Sn_{0.5}W_{0.5}S_2$ and $SnS_2$ before and after contact. $E_F$, $E_{CB}$, and $E_{VB}$ denote Fermi level, conduction band and valence band, respectively. *I–V* curves measured with TUNA for **d** a $SnS_2$ nanoplate and **e** a $Sn_{0.5}W_{0.5}S_2/SnS_2$ heterostructure, under a constant force and an applied bias voltage that was linearly ramped down

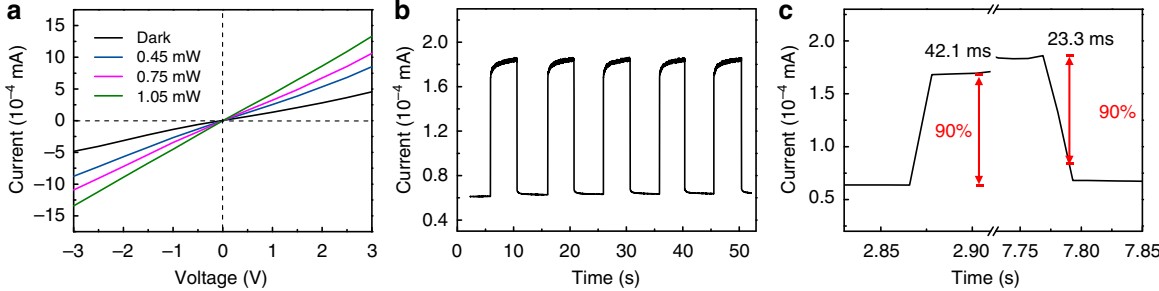

**Fig. 6** Photodetector performance of $Sn_{0.5}W_{0.5}S_2/SnS_2$ heterostructures. **a** *I–V* curves at different light intensity, **b** temporal photocurrent response and **c** a zoom-in view of the temporal photocurrent response of a photodetector based on $Sn_{0.5}W_{0.5}S_2/SnS_2$ heterostructures. The light source used for all measurements was a 405 nm laser

showed the best sensitivity (i.e., sensing response, $\Delta R/R_0$) at 100 ppb levels among all reported chemiresistive sensors operating at room temperature (Supplementary Table 3). Furthermore, as shown in Supplementary Fig. 22, a typical $Sn_{0.5}W_{0.5}S_2/SnS_2$ sensor was repeatedly exposed to 1 ppm acetone and then back to $N_2$ gas for 10 cycles and showed an almost constant sensing response of $1.88 \pm 0.07\%$ (by taking standard deviation of the results from 10 cycles), indicating its good repeatability.

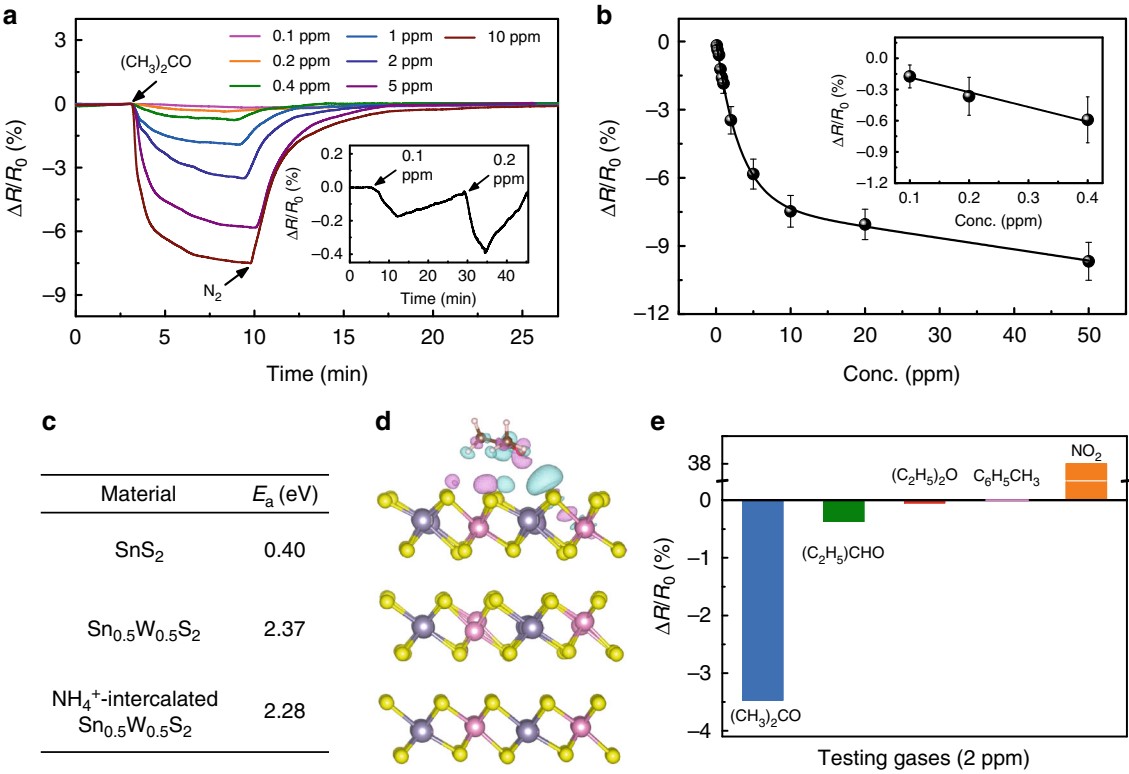

**Fig. 7** Gas sensing performance of $Sn_{0.5}W_{0.5}S_2/SnS_2$ heterostructures. **a** Response–recovery curves of a typical chemiresistive sensor fabricated from $Sn_{0.5}W_{0.5}S_2/SnS_2$ heterostructures in response to acetone gas with increasing concentrations. Inset: zoom-in response of the sensor towards 0.1 and 0.2 ppm acetone. **b** Normalized change of resistance of $Sn_{0.5}W_{0.5}S_2/SnS_2$ sensors at various acetone concentrations. Inset: zoom-in normalized change of resistance at low acetone concentrations. Each error bar indicates the standard deviation of the change of resistance for 5 experimental replicates. **c** Calculated adsorption energy, $E_a$ (eV), of acetone on different sensing materials. **d** Side view of the fully relaxed structural model of $Sn_{0.5}W_{0.5}S_2$ with surface adsorption of an acetone molecule. Cyan regions indicate charge accumulation, while pink regions represent charge depletion. **e** Comparison of the responses of the sensor towards different gases, including acetone, diethyl ether, propanal, toluene, and $NO_2$

The improved sensitivity after deposition of $Sn_{0.5}W_{0.5}S_2$ nanosheets on $SnS_2$ nanoplates could be attributed to the following reasons. First, the formation of an ohmic-like contact between $Sn_{0.5}W_{0.5}S_2$ and $SnS_2$ that allowed rapid charge transfer across the metal–semiconductor interface partly led to a 35 times reduction in background noise and thus a much higher signal to noise (S/N) ratio (Supplementary Fig. 23)[27]. Second, it can be noted that, the response/recovery time of the $Sn_{0.5}W_{0.5}S_2/SnS_2$ gas sensor was longer than that of the $SnS_2$ gas sensor (Fig. 7a and Supplementary Fig. 21), pointing to a chemical adsorption-related sensing pathway[64,65]. To confirm this, we calculated the adsorption energy of acetone on the different sensing materials (Fig. 7c). The adsorption energy of acetone on $Sn_{0.5}W_{0.5}S_2$ is 2.37 eV is much larger compared to that on $SnS_2$ (0.40 eV). This indicates that acetone molecules interact more strongly with $Sn_{0.5}W_{0.5}S_2$. It is worth noting that the $NH_4^+$-intercalation did not significantly change the adsorption ability of $Sn_{0.5}W_{0.5}S_2$ nanosheets towards acetone, with the adsorption energy slightly reduced by 0.09 eV. As further shown in Fig. 7d, there is an obvious charge accumulation on $Sn_{0.5}W_{0.5}S_2$ due to charge transfer from the absorbed acetone molecule. Third, one of the advantages of using low-dimensional materials in gas sensing as compared with bulk crystals is their large specific surface areas, which are beneficial for providing large interfaces for channel–gas interaction[25]. Indeed, the deposition of wrinkled $Sn_{0.5}W_{0.5}S_2$ nanosheets on $SnS_2$ nanoplates increased the specific surface areas from typically 6.25 to 11.37 $m^2$/g based on the Brunauer–Emmett–Teller (BET) measurements as shown in

Supplementary Fig. 24. However, in spite of the beneficial effects from the large specific surface area of the wrinkled $Sn_{0.5}W_{0.5}S_2$ nanosheets and their strong interaction with acetone, increasing the amount of $Sn_{0.5}W_{0.5}S_2$ nanosheets on $SnS_2$ did not further improve the sensing performance, but on the contrary, led to poorer sensitivity with a minimal detectable concentration of only 1 ppm (Supplementary Fig. 25). This suggests that the concentration of the metallic phase present in the hybrid sensing film should not be too high, otherwise, the gas-induced doping effect on the semiconducting $SnS_2$ would be substantially weakened. Therefore, the presence of the semiconductor/metal heterostructures with a low charge transfer barrier, combined with sufficient active surfaces for strong gas adsorption are important in achieving low sensitivity in our thin film based gas sensors.

The selectivity of the $Sn_{0.5}W_{0.5}S_2/SnS_2$ sensor was also investigated by comparing its sensing response towards acetone with other electron donating gases like diethyl ether and propanal, neutral gas like toluene, and electron withdrawing gas like $NO_2$ (Fig. 7e, Supplementary Fig. 26). The response of the $Sn_{0.5}W_{0.5}S_2/SnS_2$ sensor towards diethyl ether and propanal was 10 times lower compared to that towards acetone at 2 ppm. This is largely due to the weaker electron donating ability of diethyl ether and propanal compared to that of acetone (Fig. 7e). Under the exposure of toluene, the sensor showed no response. In sharp contrast, when responding to $NO_2$, an electron withdrawing gas, the sensor showed increased resistance (Fig. 7e, Supplementary Fig. 26e), agreeing with the fact that $SnS_2$ is an n-type semiconductor (Supplementary Fig. 18b).

## Discussion

The epitaxial growth of metallic $Sn_{0.5}W_{0.5}S_2$ nanosheets on the surfaces of $SnS_2$ nanoplates were realized via a solution-phase epitaxial deposition process. Importantly, an alloyed metal oxide (i.e., $Sn_{0.17}WO_3$) was identified as an intermediate product which formed at 180 °C and decomposed subsequently at 220 °C to provide additional Sn and W ions with a high W/Sn ratio of about 30. We proposed that this high W/Sn concentration could drive the formation of alloyed $Sn_{0.5}W_{0.5}S_2$ nanosheets despite the fact that the formation of Sn–S bond requires less energy compared to that of the W–S bond. Although $1T-SnS_2$ is a semiconductor, alloyed $Sn_{0.5}W_{0.5}S_2$ showed 83% distorted 1T structure, which is metal-like as predicted by theoretical calculations. KPFM and TUNA measurements on a $Sn_{0.5}W_{0.5}S_2/SnS_2$ heterostructure suggested the formation of ohmic-like contact at the heterointerface, resulting in a low charge transfer resistance. The rapid charge transport at the $Sn_{0.5}W_{0.5}S_2/SnS_2$ heterointerface allowed for the fabrication of fast photodetector with a short response time of 42 ms. Additionally, when the $Sn_{0.5}W_{0.5}S_2/SnS_2$ heterostructures were fabricated into thin films for gas sensing, a much enhanced signal-to-noise ratio was achieved partly due to the presence of metallic $Sn_{0.5}W_{0.5}S_2$ layers. Furthermore, $Sn_{0.5}W_{0.5}S_2$ showed much enhanced surface adsorption of acetone than $SnS_2$ based on theoretical calculations. As a consequence, selective and sensitive detection of acetone was achieved with an ultralow minimum detectable concentration of 100 ppb at room temperature. This, combined with large sensing responses at 100 ppb levels outperforms previously reported room-temperature chemiresistive sensors. Our use of semiconducting $SnS_2$ nanoplates as synthesis templates for the solution-phase epitaxial growth of metallic $Sn_{0.5}W_{0.5}S_2$ nanosheets demonstrates a promising way towards the facile, economic and high-yield preparation of functional hybrid nanomaterials. Synthesis of layered materials through alloying among elements with distinct electronic and chemical properties is expected to bring about more materials and unusual phenomena.

## Methods

**Materials**. Ammonium tungstate hydrate $((NH_4)_{10}H_2(W_2O_7)_6$, 99.99%), ammonium molybdate tetrahydrate $((NH_4)_6Mo_7O_{24}\cdot4H_2O$, 99.0%), thiourea $(CS(NH_2)_2$, 99.0%) and tin tetrachloride hydrate $(SnCl_4\cdot5H_2O$, 99.9%) were purchased from Sigma-Aldrich (Shanghai, China). Ethanol $(C_2H_5OH$, ACS, 99.9%) was purchased from J&K chemical (Shanghai, China). The gaseous analytes $(CH_3COCH_3$, $(CH_3CH_2)_2O$, $CH_3CH_2CHO$, $C_6H_5CH_3$, and $NO_2)$ which were diluted with $N_2$ gas at concentrations of 1000 ppm were purchased from Nanjing Teqi Co., Ltd. All chemicals were used as received without further purification. The deionized (DI) water was purified using Milli-Q3 System (Millipore, France).

**Preparation of $SnS_2$ nanoplates**. In a typical process, 0.25 mmol of $SnCl_4\cdot5H_2O$ and 3.75 mmol of $CS(NH_2)_2$ were dissolved in 19.45 mL DI water and stirred for 2 h to form a homogeneous solution. This solution was transferred to a 25 mL Teflon-lined stainless steel autoclave, heated to 220 °C in an electrical oven and then maintained at this temperature for 12 h before being cooled down naturally to room temperature. The obtained product was centrifuged at 8000 rpm for 10 min, and the precipitate was washed with DI water for three times before further characterization.

**Preparation of $Sn_{0.5}W_{0.5}S_2/SnS_2$ heterostructures**. In a typical process, 0.25 mmol of $(NH_4)_{10}H_2(W_2O_7)_6$, 7.5 mmol of $CS(NH_2)_2$, and 0.5–0.625 mmol of $SnCl_4\cdot5H_2O$ were dissolved in 19.45 mL DI water and stirred at 80 °C for 2 h to form a homogeneous solution. This solution was then transferred to a 25 mL Teflon-lined stainless steel autoclave, heated to 220 °C in an electrical oven and then maintained at this temperature for 60 h before being cooled down naturally to room temperature. The obtained product was centrifuged at 8000 rpm for 10 min, and the precipitate was washed with DI water for three times before further characterization.

**Characterizations**. Scanning electron microscope (SEM, JEOL JSM-7800F, Japan), transmission electron microscope (TEM, JEOL 2100Plus, Japan) and high resolution transmission electron microscope (HRTEM, JEOL 2100 F, Japan) coupled with energy dispersive X-ray (EDX) spectroscope were used to investigate the

compositional, morphological and structural features of the samples. X-ray diffraction (XRD, Rigaku SmartLab, Japan) was performed using CuKα radiation ($\lambda$ = 1.54 Å). X-ray photoelectron spectroscopy (XPS, PHI 5000 VersaProbe, Japan) measurements were conducted on the different metal sulfide nanostructures, and the binding energies were corrected for specimen charging effects using the C 1 s level at 284.6 eV as the reference. Raman spectra (Horiba HR800, France) of the samples were collected with a 532 nm laser. Semiconductor parameter analyzer (Tektronix Keithley 4200, America) and probe station (Lake Shore TTPX, America) were used to investigate the semiconductor properties of the samples. A commercial atomic force microscope (AFM, Dimension ICON with Nanoscope V controller, Bruker) was used to investigate the electrical properties of the individual nanostructures in air. Inductively coupled plasma mass spectrometry (ICP-MS, Agilent 7700×, America) was used to measure the concentration of Sn and W ions in the synthesis solution. Brunauer–Emmett–Teller (BET, Micromeritics, ASAP2460, USA) measurements were carried out to determine the specific surface area and pore size distribution of various samples.

**Semiconducting property characterization**. After Au (50 nm)/Cr (30 nm) drain and source electrodes were deposited onto a $SiO_2$ (285 nm)/Si substrate via thermal evaporation through a shadow mask, $SnS_2$ nanoplates or $Sn_{0.5}W_{0.5}S_2/SnS_2$ heterostructures in water were drop-casted onto the substrate, acting as the channel to connect the drain and source electrodes with a channel length of 11 μm. The semiconducting properties of the channel materials were then characterized using a Keithley 4200 semiconductor characterization system operating at 77 K in vacuum ($5 \times 10^{-5}$ Torr).

**KPFM and TUNA measurements**. After Au (50 nm)/Cr (30 nm) was coated onto a $SiO_2$ (285 nm)/Si substrate via thermal evaporation, $SnS_2$ nanoplates (or $Sn_{0.5}W_{0.5}S_2/SnS_2$ heterostructures) in water were drop-casted onto the substrate. A KPFM (Dimension ICON with Nanoscope V controller, Bruker) was then used to characterize the surface potential of the $SnS_2$ nanoplates (or $Sn_{0.5}W_{0.5}S_2/SnS_2$ heterostructures) at ambient conditions. The contact potential difference between the tip (PtIr) and the sample surface ($V_{CPD}$), which is also referred to as the surface potential can be calculated by using the following equations:

$$V_{CPD} = \frac{1}{e}\left(\varphi_t - \varphi_f\right) \qquad (1)$$

$$\Delta V_{CPD} = \Delta V_{CPD}(film) - \Delta V_{CPD}(substrate)$$

$$\Delta V_{CPD} = \frac{1}{e}\left(\varphi_t - \varphi_f\right) - \frac{1}{e}\left(\varphi_t - \varphi_s\right) = \frac{1}{e}\left(\varphi_s - \varphi_f\right) \qquad (2)$$

where $\varphi_t$, $\varphi_s$, and $\varphi_f$ represent the work functions of the probe tip, the substrate, and the sample film, respectively.

PeakForce Tunneling atomic force microscopy (TUNA, Dimension ICON with Nanoscope V controller, Bruker) was used to investigate the current–voltage (I–V) characteristics of individual $SnS_2$ nanoplates or $Sn_{0.5}W_{0.5}S_2/SnS_2$ heterostructures. During the measurement, the PtIr tip was pressed against the sample with a constant force, feedback was switched to contact mode, and the voltage was linearly ramped up and down while the current signal was collected. Analysis of the I–V curves was performed with the Nanoscope Analysis software.

**Fabrication of photodetector and photoresponse measurements**. Au (15 nm in thickness) interdigitated electrodes with a 10 nm spacing were deposited onto a $SiO_2$ (300 nm)/Si substrate via magnetron sputtering through a shadow mask. After that, 0.5 μL of a concentrated dispersion (6.0 mg/mL) of the $Sn_{0.5}W_{0.5}S_2/SnS_2$ heterostructures or $SnS_2$ nanoplates was drop-casted on the electrodes.

The current–voltage (I–V) and the current–time (I–t) curves of the photodetectors were measured on a semiconductor characterization system (Keithley 4200, USA) in air at room temperature. A 405 nm laser was used for all the measurements. The actual power intensity was measured by a power meter (LP1, Sanwa Electric Instrument Co., Ltd., Japan).

**Fabrication of chemiresistive sensors and gas sensing tests**. Chemiresistive gas sensors were fabricated based on $SnS_2$ nanoplates or $Sn_{0.5}W_{0.5}S_2/SnS_2$ heterostructures for sensing of various gases, including $CH_3COCH_3$, $(CH_3CH_2)_2O$, $CH_3CH_2CHO$, $C_6H_5CH_3$, and $NO_2$. Typically, a drop of 100 μL aqueous solution containing 10 mM as-prepared $SnS_2$ or $Sn_{0.5}W_{0.5}S_2/SnS_2$ was drop-casted onto an Au interdigitated electrode (with 0.1 mm spacing over a $2 \times 1$ cm$^2$ area, Changchun Mega Borui Technology Co., Ltd) and then dried in oven at 60 °C. The gas sensing test was performed in an air-tight chamber with electrical feedthroughs at room temperature (25 °C). A constant current was applied to the sensor electrode, and the variation of the sensor resistance was monitored and recorded with the changes in the gas environment using a data acquisition system (34972A, Agilent) with a 20 channel multiplexer (34901A, Agilent). A typical sensing measurement cycle consisted of three sequential steps: (1) a dried $N_2$ flow was introduced into the chamber to record a baseline resistance ($R_0$); (2) a target gas, e.g., acetone, balanced in $N_2$ was introduced, and the concentration increased (0.1–50 ppm) with

progressive cycles; and (3) when the resistance of the sensor reached equilibrium in the target gas ($R_a$), the target gas was replaced by $N_2$ to allow the resistance of the sensor to return to $R_0$. All gas flows were set at 500 sccm, precisely controlled by using mass-flow controllers.

**Computational methods**. All the computations were performed with Vienna Ab initio simulation package (VASP) which is based on the density functional theory (DFT)[66,67]. The exchange-correlation interaction uses the general gradient approximation (GGA) formulated by Perdew–Burke–Ernzerhof (PBE)[66]. All electron interactions were described with projector augmented wave (PAW) pseudo potentials. Long-range dispersion corrections have been considered within the DFT-D2 method. The dispersion coefficient $C_6$ and van der Waals radius $R_0$ for H, C, N, O, S, Sn and W used in our DFT-D2 method were taken from previous reports[68,69]. The scale factor $S_6$ was set to 0.75 because the GGA-PBE function was employed. An $11 \times 11 \times 1$ k-point mesh was used for the interaction of the Brillouin-zone. The cutoff energy for the plane wave basis set was restricted to 400 eV, and a vacuum region of at least 12 Å was used in building the slab models. The convergence threshold was set as $10^{-4}$ eV in energy and 0.02 eV/Å in force, respectively. As shown in Fig. 4c, a four-layer 1T-$Sn_{0.5}W_{0.5}S_2$ on a monolayer 1T-$SnS_2$ was constructed with a $2 \times 2$ supercell, which contains 24 Sn, 16 W and 80 S atoms, respectively. Three $NH_4^+$ ions were introduced in the four-layer 1T-$Sn_{0.5}W_{0.5}S_2$ on a monolayer 1T-$SnS_2$ system (Supplementary Fig. 10). All atoms in the structure were fully relaxed to optimize without any restriction, and the convergence threshold was set as $10^{-4}$ eV in energy and 0.02 eV/Å in force, respectively. The optimized lattice constants and work functions ($\Phi$) were summarized in Supplementary Table 1. The experimental lattice constant ($a = 3.46$ Å) was used in DFT calculations for the in-plane periodicity of the four-layer 1T-$Sn_{0.5}W_{0.5}S_2$ on a monolayer 1T-$SnS_2$ without and with intercalated $NH_4^+$ ions. To evaluate the stability of the adsorption of an acetone molecule on a three-layer 1T-$SnS_2$ and three-layer 1T-$Sn_{0.5}W_{0.5}S_2$ without and with intercalated $NH_4^+$ ions (each system contains a $4 \times 4$ supercell), the adsorption was defined by $\triangle E_a = E_{acetone} + E_{substrate} - E_{acetone-substrate}$ (where $E_{acetone-substrate}$ is the total energy of the acetone/substrate compound systems, whereas $E_{acetone}$ and $E_{substrate}$ are the energy of the isolated acetone molecule, and the total energies of upper two-layer relaxed and bottom-layer fixed 1T-$SnS_2$ or distorted 1T-$Sn_{0.5}W_{0.5}S_2$ without or with intercalated $NH_4^+$ ions systems, respectively. VESTA was used for preparation of the structure models[70].

## Data availability

The data that support the findings of this study are available from the corresponding author on request.

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

## Acknowledgements

This research was supported by the National Natural Science Foundation of China (Grant no. 51322202 and 21373112), the Joint Research Fund for Overseas Chinese, Hong Kong and Macao Scholars (Grant no. 51528201) and the Young 1000 Talents Global Recruitment Program of China. Q.W. and X. Huang are grateful to the High Performance Computing Center of Nanjing Tech University for providing the computational resources. H.Z. thanks the support from MOE under AcRF Tier 2 (ARC 19/15, No. MOE2014-T2-2-093; MOE2015-T2-2-057; MOE2016-T2-2-103) and AcRF Tier 1 (2016-T1-001-147; 2016-T1-002-051), and NTU under Start-Up Grant (M4081296.070.500000) in Singapore. H.Z. also would like to acknowledge the Facility for Analysis, Characterization, Testing and Simulation, Nanyang Technological University, Singapore, for use of their electron microscopy facilities.

## Author contributions

X. Huang and W.H. proposed the research direction and guided the project. Xiaoshan Wang and Z.W. designed and performed the experiments. J.Z., H.L., X. Hu, J.Q., G.H., and S.L. analyzed and discussed the experimental results. Xiang Wang, Z. Zhang, J.W., Z. Zhu, Z.L., Y.L., N.W., and B.C. performed some supporting experiments. Q.W. provided theoretical calculations and analyses. S.L., Q.H., J.C., J.Y., W.Z., T.H., and H.Z. contributed to the revision of the manuscript.

## Additional information

**Competing interests:** The authors declare no competing interests.

