## [Peer Review File · Nature Communications]

Reviewers' comments:

Reviewer #1 (Remarks to the Author):

The manuscript 'Realization of vertical metal/semiconductor heterostructures via solution-phase epitaxy' describes the fabrication via hydrothermal synthesis of $\text{Sn}_{0.5}\text{W}_{0.5}\text{S}_2/\text{SnS}_2$ heterostructures for chemiresistive gas sensing. Within the manuscript the authors make three key claims; 1) the conversion of semi-conducting TMCs to metallic due to alloying; 2) ohmic contacts within their heterostructures; and 3) exceptional performance as gas sensors. The manuscript is well written and referenced and contains high quality images, having a strong potential impact, specifically regarding synthesis routes towards alloying of TMCs and band engineering. In my opinion minor revision is required prior to publication, detailed comments follow below.

1) The use of solution phase synthesis to produce transition metal nanostructures is novel, with the majority of reports of these systems arising from vapour phase growth. However, there are publications on chalcogen alloys ($\text{MoS}_{(1-x)}\text{Se}_x$) which should be referenced [Gong et.al., ACS Catalysis, 2015, 5(4), 2213] [Li et. al., ACS Appl Mater & Inter, 2016, 8(43), 29442]. Similarly heterostructures of SnS_2/SnS have been reported via solution phase synthesis [Hu et. al., Materials Research Bulletin, 2013, 48(6), 2325]. These previous works should be highlighted. However, to the best of the reviewers knowledge this is the first report of a one pot-synthesis of a alloy/single component heterostructure through solution phase synthesis.

2) The authors describe the crystal structure of their alloy and heterostructure in detail through XRD, XPS, SAED, and Raman spectroscopy. Yet despite this there remains ambiguity about the structure of observed crystal phase, as evidenced by the comment "Note that in the present study, for simplicity, we also use 1T to denote distorted-1T structures". Such a simplification is highly problematic as both commonly observed versions of distorted 1T structures (1T' and Td) in TMCs result in significant changes to the electronic structure of the material [Voiry et. al., Chem Soc Rev, 2015, 44, 2702][Duerloo et. at., Nat Comms, 2014, 5, 4214]. The authors should clarify their distorted 1T phases observed within their material.

3) The metallic property of the heterostructures was well studied via DFT, KPFM, and TFT measurements and the cause of this change was discussed. For TMCs such as WS_2 and MoS_2 the use of a NH_4^+ cation is known to be able to induce a metallic 1T phase change, this should be mentioned in the text.

4) The gas sensing performance of the heterostructure is again well described and studied in detail. However, while the performance is excellent there are other materials in the literature which are similar in performance for acetone sensing such as W-NiO at 0.1ppm levels [Wang et. al., Sensors and Actuators B: Chemical, 2015 220, 59]. The authors should add a discussion of why their material is superior or advantageous over competing systems. Further the change in resistance at 0.1 and 0.2 ppm measurements (Figure 6, S15) is very difficult to see. Due to this difficulty and the import of the low limit of detection plotting Fig 6b in a way to show the resistance change at lower gas concentrations clearly is important. Further, a statistical comment on the R/R_0 for each concentration measured would boost confidence that the 0.1ppm point can be repeatedly measured.

Reviewer #2 (Remarks to the Author):

Transition metal chalcogenides play an important role in microelectronic and optoelectronic devices due to their advantages, such as optical transmission, high conductivity and sensitivity. Nevertheless, it was widely accepted that there are different crystal phases, such as the 2H and 1T

polytypes for transition metal chalcogenides, which are of distinct electronic properties. Fortunately, the semiconductor-to-metal transition in above system can be realized by various engineering, such as doping and strain, which offer some interesting and potential application. In the manuscript, the metallic Sn_{0.5}W_{0.5}S₂ nanosheets have been in-situ synthesized and hybridized with SnS₂ nanoplates, which can form the vertical heterostructures. It was found that a good Ohmic contact can be obtained at the Sn_{0.5}W_{0.5}S₂/SnS₂ heterointerface, which shows a low charge transport resistance. Moreover, the authors constructed gas sensors based on these heterostructures, which have the highest responses to acetone among all reported room-temperature chemiresistive sensors. This work is interesting for design solution-processible thin-film devices. I suggest to publish after minor revision.

1. For WS₂, the metallic structure (1T) is unstable in the absence of external stabilizing influences. However, 1T-WS₂ has been observed in this work, which plays a crucial role in the realization of metallic Sn_{0.5}W_{0.5}S₂ nanosheets. The intrinsic mechanism for the formation of 1T WS₂ is not clear. More explanation should be given.
2. In order to match with the theoretical results, the composition of W (0.5) is critical. The authors only take XPS technique to check the composition. As we know, the XPS measurement is sensitive to surface, not for the nanostructure. Especially, the electronic state from different phase structure is the same for the elements. How can they distinguish them in Figure 2?
3. The physical mechanism of the gas sensitivity is needed to be added. The authors take some general explanations in the textbook to clarify the phenomena. However, one can take care of the transition metal chalcogenides with 2D structure, which could show different physical properties, as compare to bulk/film system. A detailed explanation behind the observations should be presented.
4. According to Fig.S4, there is a NH₄⁺ intercalation with a molar concentration of ~20 %. Does the NH₄⁺ have any influence on the behavior of gas sensors based on Sn_{0.5}W_{0.5}S₂/SnS₂ heterostructures? Please check the issue.
5. In Fig.S14 (d), the curves of I_d at V_g from -30 to 20 V are missing. Please correct it.

Reviewer #3 (Remarks to the Author):

This is an interesting paper regarding the synthesis of the metallic Sn-W binary sulphide that forms an Ohmic-like contact with SnS₂, which could be important for developing high performance electronics. However, the use of VOC gas sensing as a demonstrating model here is a bit inappropriate. Firstly, the authors neglect the effect of gas molecule adsorption of the metallic surface of Sn-W binary sulphide and ascribe the improved gas sensing performance to the facile charge transfer induced by the Ohmic-like contact. As the gas sensing mechanism is complicated and hybrid, the significance of such a heterostructure is greatly disadvantaged. Secondly, although low detection limit is achieved, the gas sensor is not practical in the reality as the response/recovery kinetics are slow and the base-line drifting effect is obvious. In addition, repeatability is not demonstrated here. Therefore, I suggest the authors should significantly re-structure and re-write this manuscript to emphasize the unique properties of this Ohmic-like heterostructure.

There are a few additional major concerns here:

1. The justification of using Sn-W binary compound is missing. Why the authors did not try other transition metal such as Mo?
2. The determination of Sn_{0.5}W_{0.5}S₂ is purely based on EDX, which can be influenced by many factors. I suggest to use XPS to further confirm it.
3. The synthesis method is chemically based and therefore statistics on the size, thickness and Sn-W ratio should be carried out based on a large number of samples.
4. The authors mentioned that the Sn_{0.5}W_{0.5}S₂ covers most of SnS₂. What are the thickness of the heterostructure and each individual component?
5. The role of intercalated NH₄⁺ ions towards gas sensing performance is missing here. Will the

NH₄ ions be active in attracting the VOC gas molecules?

6. It is noticed that the response/recovery kinetics of pure SnS₂ is much better than the heterostructure. What is the reason? It seems to me that this is contradictory to the facile charge transfer induced by the Ohmic-like contact as emphasised by the authors. It may reflect an unfavourable charge transfer pathway if it is purely based on the gas sensing performances.

Point-by-point response to referees' comments

Referee #1:

The manuscript 'Realization of vertical metal/semiconductor heterostructures via solution-phase epitaxy' describes the fabrication via hydrothermal synthesis of $\text{Sn}_{0.5}\text{W}_{0.5}\text{S}_2/\text{SnS}_2$ heterostructures for chemiresistive gas sensing. Within the manuscript the authors make three key claims; 1) the conversion of semi-conducting TMCs to metallic due to alloying; 2) ohmic contacts within their heterostructures; and 3) exceptional performance as gas sensors. The manuscript is well written and referenced and contains high quality images, having a strong potential impact, specifically regarding synthesis routes towards alloying of TMCs and band engineering. In my opinion minor revision is required prior to publication, detailed comments follow below.

Comment 1: The use of solution phase synthesis to produce transition metal nanostructures is novel, with the majority of reports of these systems arising from vapour phase growth. However, there are publications on chalcogen alloys ($\text{MoS}_{(1-x)}\text{Se}_x$) which should be referenced [Gong et. al., ACS Catalysis, 2015, 5(4), 2213] [Li et. al., ACS Appl Mater & Inter, 2016, 8(43), 29442]. Similarly heterostructures of SnS_2/SnS have been reported via solution phase synthesis [Hu et. al., Materials Research Bulletin, 2013, 48(6), 2325]. These previous works should be highlighted. However, to the best of the referees knowledge this is the first report of a one pot-synthesis of a alloy/single component heterostructure through solution phase synthesis.

Response: We thank the referee very much for the suggestions. We have cited the suggested papers in the revised manuscript.

Comment 2: The authors describe the crystal structure of their alloy and heterostructure in detail through XRD, XPS, SAED, and Raman spectroscopy. Yet despite this there remains ambiguity about the structure of observed crystal phase, as evidenced by the comment "Note that in the present study, for simplicity, we also use 1T to denote distorted-1T structures". Such a simplification is highly problematic as both commonly observed versions of distorted 1T structures (1T' and T_d) in TMCs result in significant changes to the electronic structure of the material [Voiry et. al., Chem Soc Rev, 2015, 44, 2702] [Duerloo et. at., Nat Comms, 2014, 5, 4214]. The authors should clarify their distorted 1T phases observed within their material.

Response: We thank the referee very much for the comments/suggestions. It is interesting that from STEM images (**Figure S9** below), we did not observe zigzag lattice patterns for 1T' or T_d phase. Although these phases have been observed in TMCs like WS_2 and WTe_2 (Chen et al. Nat. Commun., **2018**, 9, 2003; Fei et al. Nat. Phys., **2017**, 13, 677), they have not been reported

as possible crystal structures for SnS₂. Based on our calculation results shown in **Table S1** and **Figure S10** below, the W-S bonds tend to be shorter than the Sn-S bonds, suggesting a distortion from the perfect in-plane symmetric 1T lattice. This kind of distortion rather than forming long-ranged 1T' or T_d phase might be due to the random mixing of W and Sn atoms. We have added the above discussion in our revised manuscript.

Figure S9. (a,b) STEM images of the edge area of a typical Sn_{0.5}W_{0.5}S₂/SnS₂ heterostructure. The inset of (a) shows the contrast profile of the line marked in the image, indicating a typical metal-sulfur-sulfur-metal atomic pattern for 1T-phased structure. From the thin area near the edge, contrast difference due to the presence of Sn and W elements can be observed.

Table S1. Optimized lattice parameters, bond lengths, and work functions (Φ) of 1T-SnS₂, 1T-WS₂, and a four-layer 1T-Sn_{0.5}W_{0.5}S₂ on a monolayer 1T-SnS₂ without and with intercalated NH₄⁺ ions.

System	a (Å)	b (Å)	c (Å)	α (°)	β (°)	γ (°)	Sn-S bond length (Å)	W-S bond length (Å)	Φ (eV)
SnS ₂	3.65	3.65	23.60	90	90	120	2.58	-	5.27
WS ₂	3.15	3.15	23.83	90	90	120	-	2.42	6.62
Sn _{0.5} W _{0.5} S ₂	3.46*	3.46*	23.34	90	90	120	2.52	2.43	5.62
NH ₄ ⁺ -intercalated Sn _{0.5} W _{0.5} S ₂	3.46*	3.46*	32.54	90	90	120	2.55	2.38	2.81

* The experimental lattice constant (3.46 Å) estimated from the electron diffraction pattern was used in our calculations for the in-plane periodicity of the four-layer 1T-Sn_{0.5}W_{0.5}S₂ on a monolayer 1T-SnS₂ without and with NH₄⁺-intercalated systems.

Figure S10. Calculated structural models of (a) SnS₂, (b) 1T-WS₂, (c) a four-layer distorted 1T-Sn_{0.5}W_{0.5}S₂ on a monolayer 1T-SnS₂ and (d) a four-layer distorted 1T-Sn_{0.5}W_{0.5}S₂ on a monolayer 1T-SnS₂ with intercalated NH₄⁺ ions.

Comment 3: The metallic property of the heterostructures was well studied via DFT, KPFM, and TFT measurements and the cause of this change was discussed. For TMCs such as WS₂ and MoS₂ the use of a NH₄⁺ cation is known to be able to induce a metallic 1T phase change, this should be mentioned in the text.

Response: We thank the referee very much for this suggestion. We have added the following discussion on the NH₄⁺ cation-induced formation of 1T-phased structures in our revised manuscript.

“The formation of 1T or distorted 1T structures have been observed previously in TMCs when they were intercalated with alkali metal ions (e.g. Li⁺, K⁺ etc.) (Eda et al. ACS Nano, **2012**, 6, 7311; Zak et al. J. Am. Chem. Soc., **2002**, 124, 4747) or synthesized in presence of ammonium containing precursors (Kappera et al. Nat. Mater., **2014**, 13, 1128; Liu et al. Adv. Mater., **2015**, 27, 4837) and hydrazine hydrate (Li et al. Nat. Commun., **2017**, 8, 15377). According to previous theoretical calculations, the presence of the positive counterions could cause an increase of the electron density of the *d*-orbital of the transition metals, leading to the stabilization of the 1T or distorted 1T phase (Wang et al. J. Am. Chem. Soc., **2014**, 136, 6693; Cheng et al. ACS Nano, **2014**, 8, 11447). Therefore, the realization of the distorted 1T-Sn_{0.5}W_{0.5}S₂ in our present work might also be a result of the intercalated NH₄⁺ cations from CS(NH₂)₂ and (NH₄)₁₀H₂(W₂O₇)₆ used in the synthesis solution.”

Comment 4: The gas sensing performance of the heterostructure is again well described and studied in detail. However, while the performance is excellent there are other materials in the literature which are similar in performance for acetone sensing such as W-NiO at 0.1 ppm levels [Wang et. al., *Sensors and Actuators B: Chemical*, **2015** 220, 59]. The authors should add a discussion of why their material is superior or advantageous over competing systems. Further the change in resistance at 0.1 and 0.2 ppm measurements (Figure 6, S15) is very difficult to see. Due to this difficulty and the import of the low limit of detection plotting Fig 6b in a way to show the resistance change at lower gas concentrations clearly is important. Further, a statistical comment on the R/R_0 for each concentration measured would boost confidence that the 0.1 ppm point can be repeatedly measured.

Response: We thank the referee very much for the comments/suggestions. Although some metal oxide based acetone sensors, such as the one mentioned by the referee (Wang et. al., *Sensors and Actuators B: Chemical*, **2015**, 220, 59), could achieve a detection limit down to 0.1 ppm, they operated at elevated temperatures, e.g. 250 °C. A high operation temperature can increase the electrical conductivity of the sensing materials and facilitate redox reactions of acetone on their surfaces. However, it is unfavorable for device miniaturization and power consumption. In contrast, our reported $\text{Sn}_{0.5}\text{W}_{0.5}\text{S}_2/\text{SnS}_2$ heterostructure based sensor can operate at room temperature (25 °C) which is more favorable for practical use.

As for **Figure 7a-b** and **Figure S20** in the revised manuscript, we have obtained better results and provided clearer graphs to show the response changes at 0.1 and 0.2 ppm (**Figure 7a** below). We have also performed repeatability tests at various concentrations, and error bars have been added in **Figure 7b** as shown below.

Figure 7. (a) Response-recovery curves of a typical chemiresistive sensor fabricated from $\text{Sn}_{0.5}\text{W}_{0.5}\text{S}_2/\text{SnS}_2$ heterostructures in response to acetone gas with increasing concentrations. Inset: zoom-in response of the sensor towards 0.1 and 0.2 ppm acetone. (b) Normalized change

of resistance of $\text{Sn}_{0.5}\text{W}_{0.5}\text{S}_2/\text{SnS}_2$ sensors at various acetone concentrations. Inset: zoom-in normalized change of resistance of at low acetone concentrations.

Figure S20. Response-recovery curve of a typical $\text{Sn}_{0.5}\text{W}_{0.5}\text{S}_2/\text{SnS}_2$ -based sensor in response to acetone with different concentrations from 0.1 to 50 ppm.

Referee #2:

Transition metal chalcogenides play an important role in microelectronic and optoelectronic devices due to their advantages, such as optical transmission, high conductivity and sensitivity. Nevertheless, it was widely accepted that there are different crystal phases, such as the 2H and 1T polytypes for transition metal chalcogenides, which are of distinct electronic properties. Fortunately, the semiconductor-to-metal transition in above system can be realized by various engineering, such as doping and strain, which offer some interesting and potential application. In the manuscript, the metallic $\text{Sn}_{0.5}\text{W}_{0.5}\text{S}_2$ nanosheets have been in-situ synthesized and hybridized with SnS_2 nanoplates, which can form the vertical heterostructures. It was found that a good Ohmic contact can be obtained at the $\text{Sn}_{0.5}\text{W}_{0.5}\text{S}_2/\text{SnS}_2$ heterointerface, which shows a low charge transport resistance. Moreover, the authors constructed gas sensors based on these heterostructures, which have the highest responses to acetone among all reported room-temperature chemiresistive sensors. This work is interesting for design solution-processible thin-film devices. I suggest to publish after minor revision.

Comment 1: For WS_2 , the metallic structure (1T) is unstable in the absence of external stabilizing influences. However, 1T- WS_2 has been observed in this work, which plays a crucial role in the realization of metallic $\text{Sn}_{0.5}\text{W}_{0.5}\text{S}_2$ nanosheets. The intrinsic mechanism for the formation of 1T WS_2 is not clear. More explanation should be given.

Response: We thank the referee very much for the comments/suggestions. We have added the following discussion on the possible NH_4^+ cation-induced formation of 1T phase structures in our revised manuscript.

“The formation of 1T or distorted 1T structures have been observed previously in TMCs when they were intercalated with alkali metal ions (e.g. Li^+ , K^+ etc.) (Eda et al. ACS Nano, **2012**, 6: 7311; Zak et al. J. Am. Chem. Soc., **2002**, 124, 4747) or synthesized in presence of ammonium containing precursors (Kappera et al. Nat. Mater., **2014**, 13, 1128; Liu et al. Adv. Mater., **2015**, 27, 4837;) and hydrazine hydrate (Li et al. Nat. Commun., **2017**, 8, 15377). According to several theoretical calculations, the presence of the positive counterions could cause an increase of the electron density of the *d*-orbital of the transition metals, leading to the stabilization of the 1T or distorted 1T phase (Wang et al. J. Am. Chem. Soc., **2014**, 136, 6693; Cheng et al. ACS Nano, **2014**, 8, 11447). Therefore, the realization of the distorted 1T- $\text{Sn}_{0.5}\text{W}_{0.5}\text{S}_2$ in our present work might also be a result of the intercalated NH_4^+ cations from $\text{CS}(\text{NH}_2)_2$ and $(\text{NH}_4)_{10}\text{H}_2(\text{W}_2\text{O}_7)_6$ used in the synthesis solutions.”

Comment 2: In order to match with the theoretical results, the composition of W (0.5) is critical. The authors only take XPS technique to check the composition. As we know, the XPS measurement is sensitive to surface, not for the nanostructure. Especially, the electronic state from different phase structure is the same for the elements. How can they distinguish them in Figure 2?

Response: We thank the referee very much for the comment/question. We agree with the referee that XPS was not very reliable in determining the composition of the alloyed $\text{Sn}_{1-x}\text{W}_x\text{S}_2$ layers, as XPS is sensitive only to the surface of the deposited sample film. Another reason is that the $\text{Sn}_{1-x}\text{W}_x\text{S}_2$ nanosheets were coated on the top and bottom surfaces of a SnS_2 nanoplate, which cannot be selectively removed. Therefore, we used EDX spot analysis to determine the composition of the $\text{Sn}_{1-x}\text{W}_x\text{S}_2$ sheets at the periphery as shown in **Figure S5** below. We took an average of the results from several samples, and *x* in $\text{Sn}_{1-x}\text{W}_x\text{S}_2$ was determined to be ~ 0.5 , indicating $\text{Sn}_{0.5}\text{W}_{0.5}\text{S}_2$ was obtained.

Figure S5. (a) Distribution of x values in $\text{Sn}_{1-x}\text{W}_x\text{S}_2$ based on EDX spot analyses on the edges of several $\text{Sn}_{1-x}\text{W}_x\text{S}_2$. (b-e) Four examples of STEM images and EDX spot analyses on the edges of $\text{Sn}_{1-x}\text{W}_x\text{S}_2$ nanosheets, where the positions of the analyzed spots are highlighted in white crosses.

As for using XPS to determine the concentration of 1T and 2H phases, it has been reported that the XPS band positions of a metal element are sensitive to their oxidation states, coordination geometries and Fermi levels (Mahler et al. *J. Am. Chem. Soc.*, **2014**, 136, 14121; Voiry et al. *Chem. Soc. Rev.*, **2015**, 44, 2702; Papageorgopoulos et al. *Surf. Sci.*, **1995**, 338, 83). For example, in the W 4f spectrum of WS_2 , the doublet 1T peaks (~ 32.1 and ~ 34.2 eV) are downshifted by ~ 0.6 eV relative to the 2H doublet peaks (~ 32.7 and ~ 34.8 eV). The deconvolution of the W 4f bands have thus enabled the quantitative estimation of the 1T and 2H phase concentrations (Acerce et al. *Nat. Nanotechnol.*, **2015**, 10, 313). We have added the above explanation in our revised manuscript.

Comment 3: The physical mechanism of the gas sensitivity is needed to be added. The authors take some general explanations in the textbook to clarify the phenomena. However, one can take care of the transition metal chalcogenides with 2D structure, which could show different physical properties, as compare to bulk/film system. A detailed explanation behind the observations should be presented.

Response: We thank the referee very much for the comment and suggestion. To better understand the observed phenomena, we have carried out additional measurements and theoretical calculations. A detailed explanation is as follows.

First, because the gas sensing process can be influenced by many physical/chemical factors, to eliminate the surface chemistry effect, the facile charge transfer across the Ohmic-like hetero-interfaces was firstly examined by fabrication of a thin film photodetector based on the $\text{Sn}_{0.5}\text{W}_{0.5}\text{S}_2/\text{SnS}_2$ heterostructures, which showed ~ 50 times faster photoresponse compared to the device based on SnS_2 only (**Figure 6 and S19** as shown below). Such markedly shortened response time suggests the rapid transport of charge carriers across the $\text{Sn}_{0.5}\text{W}_{0.5}\text{S}_2/\text{SnS}_2$ heterointerfaces (Ma et al. *Adv. Mater.*, **2016**, 28, 3683; X. Song et al. *ACS Appl. Mater. Interfaces*, **2018**, 10, 2801).

Figure 6. (a) I-V curves at different light intensity, (b) temporal photocurrent response and (c) a zoom-in view of the temporal photocurrent response of a photodetector based on $\text{Sn}_{0.5}\text{W}_{0.5}\text{S}_2/\text{SnS}_2$ heterostructures. The light source used for all measurements was a 405 nm laser.

Figure S19. (a) I-V curves, (b) temporal photocurrent response and (c) a zoom-in view of the temporal photocurrent response of a photodetector based on SnS_2 nanoplates. The light source used for all measurements was a 405 nm laser.

When such $\text{Sn}_{0.5}\text{W}_{0.5}\text{S}_2/\text{SnS}_2$ heterostructures with facile charge transfer at the metal/semiconductor interfaces were used for gas sensing, a significant decrease of channel resistance from 10^7 to $10^5 \Omega$ was obtained, which partly led to a 35 times reduction in background noise as well as a much higher signal to noise (S/N) ratio (**Figure S23** as shown below).

Figure S23. The signal to noise ratio of $\text{Sn}_{0.5}\text{W}_{0.5}\text{S}_2/\text{SnS}_2$ (a) and SnS_2 (b) based gas sensor upon exposure to 2 ppm acetone.

Second, we note that the response/recovery time of the $\text{Sn}_{0.5}\text{W}_{0.5}\text{S}_2/\text{SnS}_2$ gas sensor was longer than that of the SnS_2 gas sensor, pointing to a chemical adsorption related sensing pathway (Calvi et al. *Sensors*, **2016**, 16, 731; Kim et al., *Nano Lett.*, **2014**, 14, 5941). To explain this phenomenon, we calculated the adsorption energies of acetone on the different sensing materials (**Figure 7c** as shown below). The adsorption energy of acetone on $\text{Sn}_{0.5}\text{W}_{0.5}\text{S}_2$ is 2.37 eV, much larger than that on SnS_2 (0.36 eV), indicating that acetone molecules interact more strongly with $\text{Sn}_{0.5}\text{W}_{0.5}\text{S}_2$ than with SnS_2 . It is worth noting that the NH_4^+ -intercalation caused the $\text{Sn}_{0.5}\text{W}_{0.5}\text{S}_2$ nanosheets to become slightly less attractive towards acetone with the adsorption energy reduced by 0.11 eV. As further shown in **Figure 7d**, there is an obvious charge accumulation on $\text{Sn}_{0.5}\text{W}_{0.5}\text{S}_2$ due to electron transfer from absorbed acetone molecule.

Figure 7c,d. (c) Calculated adsorption energy, E_a (eV), of acetone on different sensing materials. (d) Side view of the fully relaxed structural model of the $\text{Sn}_{0.5}\text{W}_{0.5}\text{S}_2$ with acetone adsorption. Cyan regions indicate charge accumulation, while pink regions represent charge depletion.

Third, as pointed out by the referee, one of the advantages of using low-dimensional materials in gas sensing as compared with bulk counterpart is their large specific surface areas, which are beneficial for providing large interfaces for channel-gas interaction (Kim et al. *Acc. Chem. Res.*, **2017**, 50, 1587). Evidently, after the SnS_2 nanoplates were coated with wrinkled

$\text{Sn}_{0.5}\text{W}_{0.5}\text{S}_2$ nanosheets, the specific surface area increased from typically $6.25 \text{ m}^2/\text{g}$ to $11.37 \text{ m}^2/\text{g}$ based on BET measurements as shown in **Figure S24** below.

Figure S24 (a) N_2 adsorption-desorption isotherms, and (b) DFT pore size distribution plots for SnS_2 nanoplates and $\text{Sn}_{0.5}\text{W}_{0.5}\text{S}_2/\text{SnS}_2$ heterostructures.

However, in spite of the beneficial effects from the relatively strong interaction between acetone and $\text{Sn}_{0.5}\text{W}_{0.5}\text{S}_2$ and the larger specific surface area of wrinkled $\text{Sn}_{0.5}\text{W}_{0.5}\text{S}_2$ nanosheets, increasing the amount of $\text{Sn}_{0.5}\text{W}_{0.5}\text{S}_2$ nanosheets deposited on SnS_2 did not further improve the sensing performance. As shown in **Figure S25** below, by changing the precursor concentrations, larger amount of $\text{Sn}_{0.5}\text{W}_{0.5}\text{S}_2$ nanosheets were deposited on SnS_2 . The resulting sensor showed a very poor sensitivity in terms of minimal detectable concentration and response (**Figure S25d**). This suggests that the amount of the metallic phase present in the hybrid sensing film should not be too high, otherwise, the gas-induced doping effect on the semiconducting SnS_2 would be significantly weakened.

Therefore, the presence of the semiconductor/metal heterostructures with a low charge transfer barrier, combined with sufficient active surfaces for strong gas adsorption is important in achieving low sensitivity in our thin film based gas sensors.

Figure S25. SEM image (a), TEM image (b) and EDX spot analysis (c) of Sn_{0.5}W_{0.5}S₂/SnS₂ heterostructures synthesized with the precursors mixed in a Sn:W:S atomic ratio of 1:6:15. Increased amount of wrinkled Sn_{0.5}W_{0.5}S₂ nanosheets were found to encapsulate SnS₂ nanoplates. (d) The response-recovery curve of Sn_{0.5}W_{0.5}S₂/SnS₂ sensor that contain a large amount of Sn_{0.5}W_{0.5}S₂ exposed to acetone gas with different concentrations from 1 to 100 ppm.

Comment 4: According to Fig.S4, there is a NH₄⁺ intercalation with a molar concentration of ~20 %. Does the NH₄⁺ have any influence on the behavior of gas sensors based on Sn_{0.5}W_{0.5}S₂/SnS₂ heterostructures? Please check the issue.

Response: We thank the referee very much for the question and comment. Based on our additional calculation results (**Figure 7c** shown above), the intercalated NH₄⁺ ions caused a slight reduction of the adsorption energy of acetone on Sn_{0.5}W_{0.5}S₂. This is because, as a positive counter-ion, NH₄⁺ can increase the electron density of the metals (or less positively charged), and thus slightly reduced the attraction towards the electron rich acetone molecules. This is also reflected in the calculated Bader charges shown below.

Figure (4c)(S17a). Optimized crystal structure with calculated Bader charges for a four-layer $\text{Sn}_{0.5}\text{W}_{0.5}\text{S}_2$ on a monolayer 1T- SnS_2 (4c) without and (S17a) with intercalated NH_4^+ ions.

Comment 5: In Fig.S14 (d), the curves of I_d at V_g from -30 to 20 V are missing. Please correct it.

Response: We thank the referee very much for this comment. For the thin film transistor fabricated from $\text{Sn}_{0.5}\text{W}_{0.5}\text{S}_2/\text{SnS}_2$ heterostructures, I_d did not change much with varied V_g because of the presence of metallic $\text{Sn}_{0.5}\text{W}_{0.5}\text{S}_2$, and therefore the I-V curves shown in **Figure S18d** almost overlapped.

Referee #3:

This is an interesting paper regarding the synthesis of the metallic Sn-W binary sulphide that forms an Ohmic-like contact with SnS_2 , which could be important for developing high performance electronics.

Comment 1: However, the use of VOC gas sensing as a demonstrating model here is a bit inappropriate. Firstly, the authors neglect the effect of gas molecule adsorption of the metallic surface of Sn-W binary sulphide and ascribe the improved gas sensing performance to the facile charge transfer induced by the Ohmic-like contact. As the gas sensing mechanism is complicated and hybrid, the significance of such a heterostructure is greatly disadvantaged. Secondly, although low detection limit is achieved, the gas sensor is not practical in the reality as the response/recovery kinetics are slow and the base-line drifting effect is obvious. In addition, repeatability is not demonstrated here. Therefore, I suggest the authors should

significantly re-structure and re-write this manuscript to emphasize the unique properties of this Ohmic-like heterostructure.

Response: We thank the referee very much for the comments and suggestions. During our revision, we have significantly re-structured and re-written the manuscript by including additional experiment results and theoretical calculations.

First, to emphasize the unique properties of Ohmic-like heterostructures, $\text{Sn}_{0.5}\text{W}_{0.5}\text{S}_2/\text{SnS}_2$ hybrid materials were fabricated into thin-film photodetectors, which showed ~50 times faster photoresponse compared with the device based on SnS_2 only. The following paragraph and figures have been added in the revised manuscript:

“The advantage of the facile charge transport across the Ohmic-like heterointerfaces was demonstrated in fabrication of thin film photodetectors based on the $\text{Sn}_{0.5}\text{W}_{0.5}\text{S}_2/\text{SnS}_2$ heterostructures. Fig. 6a shows the I-V curves of the device under 405 nm laser illumination with power intensity varied from 0.45 to 1.05 mW. A clear rise of the photocurrent with increasing illumination intensity was observed, indicating the effective conversion of photon flux to photogenerated carriers. In addition, the $\text{Sn}_{0.5}\text{W}_{0.5}\text{S}_2/\text{SnS}_2$ photodetector showed symmetric and linear I-V plots, which are in sharp contrast to the non-linear I-V curves observed for SnS_2 based device (Supplementary Fig. S19). This further indicates the low resistance contact formed between the semiconducting and metallic components in $\text{Sn}_{0.5}\text{W}_{0.5}\text{S}_2/\text{SnS}_2$. The temporal photoresponse of the photodetectors was measured as well as shown in Fig. 6b,c and Supplementary Fig. S19b,c. The $\text{Sn}_{0.5}\text{W}_{0.5}\text{S}_2/\text{SnS}_2$ photodetector showed an abrupt rise of photocurrent with a small response time of 42.1 ms, which is comparable and outperforms some previously reported TMC photodetectors (Lopez-Sanchez et al., Nat. Nanotechnol. **2013**, 8, 497; Shim et al. Adv. Mater. **2016**, 28, 6985). This value is also ~50 times shorter than that of the SnS_2 based photodetector (2.10 s) (Supplementary Fig. S19c). Such markedly shortened response time suggests the rapid transport of charge carriers across the $\text{Sn}_{0.5}\text{W}_{0.5}\text{S}_2/\text{SnS}_2$ heterointerfaces (Ma et al. Adv. Mater., **2016**, 28, 3683; Song et al. ACS Appl. Mater. Interfaces, **2018**, 10, 2801). Note that a relatively large dark current and thus a much reduced on/off ratio were observed for the $\text{Sn}_{0.5}\text{W}_{0.5}\text{S}_2/\text{SnS}_2$ based device as compared with the SnS_2 device. This was due to the metallic nature of the $\text{Sn}_{0.5}\text{W}_{0.5}\text{S}_2$ nanosheets. The similar phenomenon was reported previously in photodetectors based on graphene composites (Lee et al. Adv. Mater., **2015**, 27, 41; Wang et al. Adv. Mater. **2017**, 29, 1603995).”

Figure 6. (a) I-V curves at different light intensity, (b) temporal photocurrent response and (c) a zoom-in view of the temporal photocurrent response of a photodetector based on $\text{Sn}_{0.5}\text{W}_{0.5}\text{S}_2/\text{SnS}_2$ heterostructures. The light source used for all measurements was a 405 nm laser.

Figure S19. (a) I-V curves, (b) temporal photocurrent response and (c) a zoom-in view of the temporal photocurrent response of a photodetector based on SnS_2 nanoplates. The light source used for all measurements was a 405 nm laser.

Second, to address the referee's comment on the effect of gas molecule adsorption, we calculated the adsorption energies of acetone on the different sensing materials (**Figure 7c** shown below). The adsorption energy of acetone on $\text{Sn}_{0.5}\text{W}_{0.5}\text{S}_2$ is 2.37 eV, which is much larger compared to that on SnS_2 (0.36 eV), indicating that acetone molecules interact more strongly with $\text{Sn}_{0.5}\text{W}_{0.5}\text{S}_2$. It is worth noting that the adsorption energy of acetone on NH_4^+ -intercalated $\text{Sn}_{0.5}\text{W}_{0.5}\text{S}_2$ is slightly lower than that without intercalation. This suggests that NH_4^+ intercalation did not significantly change the sensing performance of $\text{Sn}_{0.5}\text{W}_{0.5}\text{S}_2$. As further shown in **Figure 7d** below, there is an obvious charge accumulation on $\text{Sn}_{0.5}\text{W}_{0.5}\text{S}_2$ due to electron transfer from the absorbed acetone. This actually explains why the response/recovery kinetics of the sensor are slow (Calvi et al. *Sensors*, **2016**, 16, 731; Kim et al. *Nano Lett.*, **2014**, 14, 5941), even though the charge transport across the Ohmic-like heterointerface is fast based on photoresponse measurements.

Figure 7c,d. (c) Calculated adsorption energy, E_a (eV), of acetone on different sensing materials. (d) Side view of the fully relaxed structural model of the $\text{Sn}_{0.5}\text{W}_{0.5}\text{S}_2$ with acetone adsorption. Cyan regions indicate charge accumulation, while pink regions represent charge depletion.

Another surface-related factor is the specific surface area. Large surface area is beneficial for providing large interfaces for channel-gas interaction (Kim et al. *Acc. Chem. Res.*, **2017**, 50, 1587). Evidently, after the SnS_2 nanoplates were coated with wrinkled $\text{Sn}_{0.5}\text{W}_{0.5}\text{S}_2$ nanosheets, the specific surface area increased from typically 6.25 to 11.37 m^2/g based on BET measurements as shown in **Figure S24** below.

Figure S24. (a) N_2 adsorption-desorption isotherms, and (b) DFT pore size distribution plots for SnS_2 nanoplates and $\text{Sn}_{0.5}\text{W}_{0.5}\text{S}_2/\text{SnS}_2$ heterostructures.

However, in spite of the beneficial effects from the strong interaction between acetone and $\text{Sn}_{0.5}\text{W}_{0.5}\text{S}_2$ and the large specific surface area of wrinkled $\text{Sn}_{0.5}\text{W}_{0.5}\text{S}_2$ nanosheets, increasing the amount of $\text{Sn}_{0.5}\text{W}_{0.5}\text{S}_2$ nanosheets deposited on SnS_2 did not further improve the sensing performance, but on the contrary, led to poorer sensitivity with a minimal detectable concentration of only 1 ppm (**Figure S25d**). This suggests that the amount of the metallic phase present in the hybrid sensing film should not be too high, otherwise, the gas-induced doping effect on the semiconducting SnS_2 could be significantly weakened.

Therefore, we propose that both the favorable surface chemistry and presence of semiconductor/metal heterostructures with a low charge transfer barrier are important in achieving low sensitivity in our thin film based gas sensors.

Figure S25. SEM image (a), TEM image (b) and EDX spot analysis (b) of $\text{Sn}_{0.5}\text{W}_{0.5}\text{S}_2/\text{SnS}_2$ heterostructures synthesized with the precursors mixed in a Sn:W:S atomic ratio of 1:6:15. Increased amount of wrinkled $\text{Sn}_{0.5}\text{W}_{0.5}\text{S}_2$ nanosheets were found to encapsulate SnS_2 nanoplates. (d) The response-recovery curve of $\text{Sn}_{0.5}\text{W}_{0.5}\text{S}_2/\text{SnS}_2$ sensor that contain a large amount of $\text{Sn}_{0.5}\text{W}_{0.5}\text{S}_2$ exposed to acetone gas with different concentrations from 1 to 100 ppm.

In addition, because the chemical gas adsorption is involved in the sensing process, the baseline drift problem can be solved by prolonging the desorption time as shown in our revised **Figure S20** below.

Figure S20. Response-recovery curve of $\text{Sn}_{0.5}\text{W}_{0.5}\text{S}_2/\text{SnS}_2$ -based sensor in response to acetone with different concentrations from 0.1 to 50 ppm.

As also suggested by the referee, we have added repeatability test result as shown in **Figure S22** below.

Figure S22. (a) Performance reproducibility of $\text{Sn}_{0.5}\text{W}_{0.5}\text{S}_2/\text{SnS}_2$ heterostructure based gas sensor upon cyclic exposure to 1 ppm acetone. (b) The corresponding responses for the tested 10 cycles.

There are a few additional major concerns here:

Comment 2: The justification of using Sn-W binary compound is missing. Why the authors did not try other transition metal such as Mo?

Response: We thank the referee very much for this question. We agree with the referee that both Mo and W are typical transition metals, whose chalcogenides show distinct crystal phase-

dependent electronic properties. We have actually tried to synthesize $\text{Sn}_{1-x}\text{Mo}_x\text{S}_2$ nanosheets on SnS_2 . As shown in **Figure R1** below, $\text{Sn}_{1-x}\text{Mo}_x\text{S}_2$ nanosheets grew epitaxially on SnS_2 , but mainly via the edge growth. This can be evidently seen from the EDX mapping of a typical $\text{Sn}_{1-x}\text{Mo}_x\text{S}_2/\text{SnS}_2$ heterostructure, in which the edge area showed a much higher Mo concentration. This phenomenon might be due the different synthesis energies required for basal growth or edge growth (Gong et al. Nat. Mater., **2014**, 13, 1135). In such a case, the obtained structures are not vertical heterostructures. This requires further investigation and we felt that including these non-vertical heterostructure results will not be beneficial to the main focus of our manuscript. However, we now have added the following sentence to clarify this in our revised manuscript:

“In our control experiments, we also tried to extend the Sn-W binary system to Sn-Mo system, but found that $\text{Sn}_{1-x}\text{Mo}_x\text{S}_2$ nanosheets grew epitaxially on SnS_2 mainly via the edge growth. This phenomenon might be due the different synthesis energies required for basal growth or edge growth (Gong et al. Nat. Mater., **2014**, 13, 1135), which requires our further investigation”.

Figure R1 (a) SEM image of as-prepared $\text{Sn}_{1-x}\text{Mo}_x\text{S}_2/\text{SnS}_2$ hybrid materials. (b) SAED pattern of a typical $\text{Sn}_{1-x}\text{Mo}_x\text{S}_2/\text{SnS}_2$ heterostructure. (c) EDX elemental mapping of a typical $\text{Sn}_{1-x}\text{Mo}_x\text{S}_2/\text{SnS}_2$ heterostructure.

Comment 3: The determination of $\text{Sn}_{0.5}\text{W}_{0.5}\text{S}_2$ is purely based on EDX, which can be influenced by many factors. I suggest to use XPS to further confirm it.

Response: We thank the referee very much for the suggestion. But because the SnS₂ component in Sn_{1-x}W_xS₂/SnS₂ heterostructures cannot be selectively removed, the Sn:W atomic ratio in Sn_{1-x}W_xS₂/SnS₂ measured with XPS was about 15:7 (**Figure R2** below). The larger Sn concentration comes from both Sn_{1-x}W_xS₂ and SnS₂ components. As suggested by the referee, we now have provided statistics on the EDX spot analyses results on several samples and took the mean value as shown in **Figure S5** below. We have revised it accordingly in our manuscript.

Figure R2. (a) XPS survey spectrum of Sn_{1-x}W_xS₂/SnS₂ heterostructures. (b) XPS analysis the composition of the Sn_{1-x}W_xS₂/SnS₂ heterostructures.

Figure S5. (a) Distribution of x values in Sn_{1-x}W_xS₂ based on EDX spot analyses on the edges of several Sn_{1-x}W_xS₂. (b-e) Four examples of STEM images and EDX spot analyses on the edges of Sn_{1-x}W_xS₂ nanosheets, where the positions of the analyzed spots are highlighted in white crosses.

Comment 4: The synthesis method is chemically based and therefore statistics on the size, thickness and Sn-W ratio should be carried out based on a large number of samples.

Response: We thank the referee very much for this suggestion. We have carried out additional measurements on the size, thickness and Sn-W ratio of the products. The typical results and statistics are shown in **Figure S1a**, **Figure S2**, **Figure 1a**, **Figure S3** and **Figure S5** below.

Figure S1a. SEM image of as-prepared SnS₂ nanoplates. Inset: lateral size distribution.

Figure S2. (a) Thickness distribution of SnS₂ nanoplates showing a mean value of ~43 nm. (b-e) Examples of AFM images and height analyses of SnS₂ nanoplates. Insets: height analysis of the region highlighted in the corresponding white rectangle.

Figure 1a. SEM image of as-prepared $\text{Sn}_{1-x}\text{W}_x\text{S}_2/\text{SnS}_2$ heterostructures. Inset: lateral size distribution.

Figure S3. (a) Thickness distribution of $\text{Sn}_{1-x}\text{W}_x\text{S}_2/\text{SnS}_2$ heterostructures, showing an average thickness of $\sim 60\ \text{nm}$. (b-e) Examples of AFM images and height analyses of $\text{Sn}_{1-x}\text{W}_x\text{S}_2/\text{SnS}_2$ heterostructures. Insets: height analysis of the region highlighted in the corresponding white rectangle.

Figure S5. (a) Distribution of x values in $\text{Sn}_{1-x}\text{W}_x\text{S}_2$ based on EDX spot analyses on the edges of several $\text{Sn}_{1-x}\text{W}_x\text{S}_2$. (b-e) Four examples of STEM images and EDX spot analyses on the edges of $\text{Sn}_{1-x}\text{W}_x\text{S}_2$ nanosheets, where the positions of the analyzed spots are highlighted in white crosses.

Comment 5: The authors mentioned that the $\text{Sn}_{0.5}\text{W}_{0.5}\text{S}_2$ covers most of SnS_2 . What are the thickness of the heterostructure and each individual component?

Response: We thank the referee very much for the question. From the side-view TEM images of typical $\text{Sn}_{0.5}\text{W}_{0.5}\text{S}_2/\text{SnS}_2$ heterostructures, the thickness of $\text{Sn}_{1-x}\text{W}_x\text{S}_2$ nanosheets are about 6~9 nm (**Figure S4** below). The average thickness of SnS_2 is about 43 nm (**Figure S2** above), and that of $\text{Sn}_{0.5}\text{W}_{0.5}\text{S}_2/\text{SnS}_2$ is about 60 nm (**Figure S3** above), consistent with the average thickness of the $\text{Sn}_{0.5}\text{W}_{0.5}\text{S}_2$ component.

Figure S4. (a-c) Side-view TEM images of typical $\text{Sn}_{1-x}\text{W}_x\text{S}_2/\text{SnS}_2$ heterostructures, revealing that the thickness of $\text{Sn}_{1-x}\text{W}_x\text{S}_2$ nanosheets grown on SnS_2 nanoplates is 6-9 nm.

Comment 6: The role of intercalated NH_4^+ ions towards gas sensing performance is missing here. Will the NH_4^+ ions be active in attracting the VOC gas molecules?

Response: We thank the referee very much for this question. Based on our calculation results, the intercalated NH_4^+ ions caused a slight reduction of the adsorption energy of acetone on $\text{Sn}_{0.5}\text{W}_{0.5}\text{S}_2$ (**Figure 7c** below). This is consistent with the fact that, as a positive counterion, NH_4^+ could increase the electron density (or less positively charged) of the metals (**Figure 4c and S17a** below), thus mildly weakened their attraction to the electron-rich acetone molecules.

Figure 7c,d. (c) Calculated adsorption energy, E_a (eV), of acetone on different sensing materials. (d) Side view of the fully relaxed structural model of the $\text{Sn}_{0.5}\text{W}_{0.5}\text{S}_2$ with acetone adsorption. Cyan regions indicate charge accumulation, while pink regions represent charge depletion.

Figure (4c)(S17a). Optimized crystal structure with calculated Bader charges for a four-layer $\text{Sn}_{0.5}\text{W}_{0.5}\text{S}_2$ on a monolayer 1T- SnS_2 (4c) without and (S17a) with intercalated NH_4^+ ions.

Comment 7: It is noticed that the response/recovery kinetics of pure SnS₂ is much better than the heterostructure. What is the reason? It seems to me that this is contradictory to the facile charge transfer induced by the Ohmic-like contact as emphasised by the authors. It may reflect an unfavourable charge transfer pathway if it is purely based on the gas sensing performances.

Response: We thank the referee very much for the question and comment. As mentioned above, the lower response/recovery kinetics of the Sn_{0.5}W_{0.5}S₂/SnS₂ gas sensor actually suggests that chemical adsorption was involved in the sensing process. As shown in **Figure 7c** above, the calculated adsorption energy of acetone on Sn_{0.5}W_{0.5}S₂ is 2.26 eV, which is much larger compared to that on SnS₂ (0.36 eV), indicating that acetone molecules interact much more strongly with Sn_{0.5}W_{0.5}S₂ than with SnS₂.

To eliminate the surface chemical effect, facile charge transport at Ohmic-like heterointerface was confirmed by measuring the photo-response of the sample thin film, which is shown in **Figure 6** mentioned above (please see our response to Comment 1). The Sn_{0.5}W_{0.5}S₂/SnS₂ photodetector showed a much abrupt rise of photocurrent with a response time of 42.1 ms, which was ~50 times shorter than that of the SnS₂ based photodetector (2.10 s). Such markedly shortened response time suggests the rapid transport of charge carriers across the Sn_{0.5}W_{0.5}S₂/SnS₂ heterointerfaces (Ma et al. Adv. Mater., **2016**, 28, 3683; Song et al. ACS Appl. Mater. Interfaces, **2018**, 10, 2801).

We highly appreciate the referees for their thorough reading and helpful comments!

REVIEWERS' COMMENTS:

Reviewer #1 (Remarks to the Author):

The authors of "Realization of vertical metal/semiconductor heterostructures" have addressed previous concerns well, which have dramatically improved the manuscript. The added clarity regarding crystal structure and device performance at low temperatures compared to literature gas sensors emphasises the potential impact of the manuscript.

Minor Comments:

1. Figure 1c) - there should be a colour scale bar for height in the AFM image.
2. Figure 2b)- there should be a scale bar for the SAED pattern (also inset of d if possible)
3. Figure 5a,b) - colour scale bar for KPFM images (NAP potential).

Reviewer #2 (Remarks to the Author):

The response letter well solved the comments, I suggest that the scripts can be accepted and published immediately

Reviewer #3 (Remarks to the Author):

It is obvious that the quality of the revised manuscript has been significantly improved. Most of my concern has also been addressed.

It seems that the introduction now becomes a little bit unbalance compared with the results. some of the key points are missing. For instance, the justification of using examples of photodetection and acetone sensing to emphasise the importance of the hetero-junctions has not been mentioned in the introduction. Therefore, I strong recommended the authors to strengthen the introduction before the paper is accepted.

Point-by-point response to referees' comments

Referee #1:

The authors of "Realization of vertical metal/semiconductor heterostructures" have addressed previous concerns well, which have dramatically improved the manuscript. The added clarity regarding crystal structure and device performance at low temperatures compared to literature gas sensors emphasises the potential impact of the manuscript.

Minor Comments:

1. Figure 1c) - there should be a colour scale bar for height in the AFM image.
2. Figure 2b)- there should be a scale bar for the SAED pattern (also inset of d if possible)
3. Figure 5a,b) - colour scale bar for KPFM images (NAP potential).

Response: We thank the referee very much for the suggestions. We have added color and scale bars in our revised manuscript.

Referee #2:

The response letter well solved the comments, I suggest that the scripts can be accepted and published immediately.

Response: We thank the referee very much.

Referee #3:

It is obvious that the quality of the revised manuscript has been significantly improved. Most of my concern has also been addressed.

It seems that the introduction now becomes a little bit unbalance compared with the results. Some of the key points are missing. For instance, the justification of using examples of photodetection and acetone sensing to emphasise the importance of the hetero-junctions has not been mentioned in the introduction. Therefore, I strong recommended the authors to strengthen the introduction before the paper is accepted.

Response: We thank the referee very much for the suggestion. We have included the following paragraph about the importance and possible applications of heterostructures in the introduction section in our revised manuscript.

“In view of their potential applications, heterostructures/heterojunctions such as InSe/graphene, MoTe₂/MoS₂, MoS₂/perovskite and graphene/MoS₂/graphene have recently shown promising performances in photodetectors, due to the improved charge separation/transport and enhanced light adsorption (Mudd et al. Adv. Mater., 2015, 27, 3760; Zhang et al. ACS Nano, 2016, 10, 3852; Yu et al. Nat. Nanotechnol., 2013, 8, 952; Zhang et al. Sci. China Mater., 2018, doi:10.1007/s40843-018-9274-y). Meanwhile, development of gas sensors for detection of volatile organic compounds (VOCs) are important in applications such as environmental monitoring and non-invasive diagnosis of diseases based on breath analysis (Konvalina et al. Acc. Chem. Res., 2014, 47, 66; Kim et al. Acc. Chem. Res., 2017, 50, 1587). Chemiresistive sensors based on metal oxides/sulfides have been used for detection of VOCs, however, high operating temperatures (typically ≥ 150 °C) are usually required to achieve good sensitivity and selectivity (Konvalina et al. Acc. Chem. Res., 2014, 47, 66; Kim et al. ACS Nano, 2016, 10, 5891). Very recently, layered materials such as SnS₂, WS₂ and Ti₃C₂T_x have demonstrated great potential for room-temperature VOC detection (Kim et al. ACS Nano, 2018, 12, 986; Mayorga-Martinez et al. Adv. Funct. Mater., 2015, 25, 5611; Giberti et al. Sensors and Actuators B, 2016, 223, 827). It is expected that creation of heterostructures may realize further improved sensing performance (Kim et al. Nat. Commun., 2014, 5, 4781; Wei et al. J. Am. Chem. Soc., 2009, 131, 17690; Yang et al. Nanoscale, 2017, 9, 5102).”